# Dual-level Adaptive Self-Labeling for Novel Class Discovery in Point Cloud Segmentation

## Abstract

We tackle the novel class discovery in point cloud segmentation, which discovers novel classes based on existing knowledge. Existing works propose an online point-wise clustering method with a simplified equal class-size constraint on the novel classes to avoid degenerate solutions. However, the inherent imbalanced distribution of novel classes in point clouds contradicts the equal class-size constraint, and point-wise clustering tends to ignore the rich spatial context information of objects, which can result in less expressive representation for semantic segmentation. To solve the above challenges, we propose a novel self-labeling strategy that adaptively generates high-quality pseudo-labels for imbalanced classes during model training. In addition, we develop a dual-level representation that incorporates regional consistency into the point-level classifier learning, reducing the noise in generated segmentation. Finally, we conduct extensive experiments on two widely used datasets, SemanticKITTI and SemanticPOSS, and the results show our method significantly outperforms the state-of-the-art by a large margin.

## 1 Introduction

Point cloud segmentation is a core problem in 3D perception (Landrieu & Simonovsky, 2018) and potentially useful for a wide range of applications, such as autonomous driving and intelligent robotics (Li et al., 2020; Roriz et al., 2021). Recently, there have been tremendous progresses in semantic segmentation of point clouds due to the utilization of deep learning techniques (Hu et al., 2020; Lai et al., 2022a). However, current segmentation methods primarily focus on a closed-world setting where all the semantic classes are known beforehand. As such it has difficulty in coping with open-world scenarios where both known and novel classes coexist, which are commonly seen in real-world applications.

For open-world perception, a desirable capability is to automatically acquire new concepts based on existing knowledge (Han et al., 2019). While there have been much effort on addressing the problem of novel class discovery for 2D or RGBD images (Nakajima et al., 2019; Han et al., 2021; Fini et al., 2021; Zhao et al., 2022), few work have explored the corresponding task for 3D point clouds. Only recently, Riz et al. (2023) propose an online point-wise clustering method for discovering novel classes in 3D point cloud segmentation. To avoid degenerate solutions, their method relies on an equal class-size constraint on the novel classes. Despite its promising results, such a simplified assumption faces two key challenges in point cloud segmentation: First, the distribution of novel classes in point clouds is inherently imbalanced due to different physical sizes of objects and density of points. Imposing equal-size constraint can be restrictive, causing splitting of large classes or merging of smaller ones. In addition, point-wise clustering tends to ignore the rich spatial context information of objects, which leads to less expressive representation for semantic segmentation.

To tackle the above challenges, we propose a dual-level adaptive self-labeling framework for novel class discovery in point cloud segmentation. The key idea of our approach is two-fold: 1) We design a novel self-labeling strategy that adaptively generates high-quality imbalanced pseudo-labels for model training, which facilitates clustering novel classes of varying sizes; 2) To incorporate semantic context, we develop a dual-level representation of 3D points by grouping points into regions and jointly learns the representations of novel classes at both the point and region levels. Such a dual-level representation imposes additional constraints on grouping of the points likely belonging to the same category, hence mitigating the noise in generated segmentation.

Specifically, our framework employs an encoder to extract point features for the input point cloud and average pooling to compute representations of pre-computed regions. Both types of features are fed into a prototype-based classifier to generate predictions across both known and novel categories for each point and region. To learn the feature encoder and class prototypes, we introduce a self-labeling-based learning procedure that iterates between pseudo-label generation for the novel classes and the full model training with cross-entropy losses on points and regions. Here the key step is to generate imbalanced pseudo labels, which is formulated as a semi-relaxed Optimal Transport (OT) problem with adaptive regularization on class distribution. Along the training, we employ a data-dependent annealing scheme to adjust the regularization strength. Such a design prevents discovering degenerate solutions and meanwhile enhances the model flexibility in learning the imbalanced data distributions.

To demonstrate the effectiveness of our approach, we conduct extensive experiments on two widely-used datasets: SemanticKITTI (Behley et al., 2019) and SemanticPOSS (Pan et al., 2020). The experimental results show that our method significantly outperforms the state of the art by a large margin. Additionally, we conduct a comprehensive ablation study to evaluate the significance of the different components of our method. The contributions of our method are summarized as follows:

1. We propose a novel adaptive self-labeling framework for novel class discovery in point cloud segmentation, capable of better modeling imbalanced novel classes.

2. We develop a dual-level representation for learning novel classes in point cloud data, which incorporates semantic context via augmenting the point prediction with regional consistency.

3. Our method achieves significant performance improvement on the SemanticPOSS and SemanticKITTI datasets across nearly all the experiment settings.

## 2 RELATED WORK

**Point cloud semantic segmentation.** Point cloud semantic segmentation has attracted much attention in recent years (Choy et al., 2019; Zhang et al., 2020; Zhu et al., 2021; Li et al., 2022). While previous methods have made significant progress, their primary focus is on closed-world scenarios that heavily rely on annotations for each class and cannot address open-world challenges. In contrast, we aim to develop a model to discover novel classes in 3D open-world scenarios. In the context of point cloud representation learning, incorporating spatial context is pivotal for enhancing representation learning. Long et al. (2023b) and Zhang et al. (2023c) introduce a hierarchical representation learning strategy that leverages regions as intermediaries to connect points and semantic clusters. Unlike them, we develop a dual-level learning strategy that concurrently learns to map points and regions to semantic classes. Thanks to the learning of region-level representation, our method is less sensitive to the local noises in point clouds. Moreover, we cluster regions into semantic classes by an imbalance-aware self-labeling algorithm instead of simple K-Means.

**Novel class discovery.** The majority of research on Novel Class Discovery (NCD) has primarily focused on learning novel visual concepts in 2D image domain via designing a variety of unsupervised losses on novel class data or regularization strategies (Han et al., 2019; Zhao & Han, 2021; Fini et al., 2021; Yang et al., 2022; Zhang et al., 2023b; Gu et al., 2023). In particular, Zhang et al. (2023a) consider the NCD task in long-tailed classification scenarios, and develop a bi-level optimization strategy for model learning. It adopts a fixed regularization to prevent degeneracy, which tends to impose strong restriction on learned representations, and a complicated dual-loop iterative procedure for optimization. In contrast, we propose an adaptive regularization strategy, which is critical for the success of our self-labeling algorithm. Moreover, our formulation leads to a convex problem for pseudo-label generation, which can be solved efficiently by a fast scaling algorithm (Cuturi, 2013; Chizat et al., 2018) (see Appendix A for detail comparison). More recently, Riz et al. (2023) generalizes the NCD for the task of point cloud semantic segmentation. Assuming a uniform distribution of novel classes, they develop an optimal-transport-based self-labeling algorithm to cluster novel classes. However, the method neglects intrinsic class imbalance and spatial context in point cloud data, often leading to sub-optimal clustering results.

**Optimal transport for pseudo labeling.** Unlike naive pseudo labeling (Lee et al., 2013), Optimal Transport (OT) (Villani et al., 2009)-based methods allow us to incorporate prior class distribution into pseudo label generation. Therefore, it has been used as a pseudo-label generation strategy for a wide range of machine learning tasks, including semi-supervised learning (Lai et al., 2022b; Tai

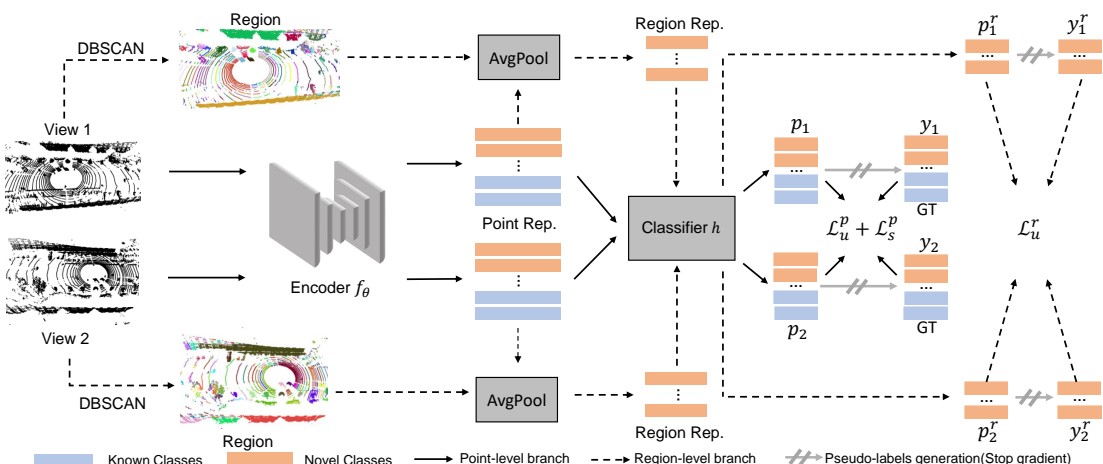

Figure 1: Our method initiates with two views of the same point cloud, clustering the points into corresponding regions. Subsequently, we extract point features through a forward pass and calculate regional representations by averaging the point features within each region. Next, predictions $p$ are made by the classifier $h$, and pseudo-labels $y$ are generated for unlabeled points and regions using our novel adaptive self-labeling algorithm. Note that generating pseudo-labels does not involve gradients. The superscript $r$ denotes the region, while subscripts 1 and 2 correspond to the two different views, and $s$, $u$ denote seen and unseen classes. Finally, we exchange the pseudo-labels between the two views and update the model accordingly. Specifically, for point-level, we compute $\mathcal{L}_u^p(y_2, p_1)$ and $\mathcal{L}_u^p(y_1, p_2)$. For region-level, we compute $\mathcal{L}_u^r(y_2^r, p_1^r)$ and $\mathcal{L}_u^r(y_1^r, p_2^r)$.

et al., 2021; Taherkhani et al., 2020), clustering (Asano et al., 2020; Caron et al., 2020), and domain adaptation (Flamary et al., 2016; Chang et al., 2022; Liu et al., 2023). However, most of these works assume the prior class distribution is either known or simply the uniform distribution, which is restrictive for NCD. By contrast, we consider a more practical scenario, where the novel class distribution is unknown and imbalanced, and design a semi-relaxed OT formulation with a novel adaptive regularization.

## 3 METHOD

In this section, we first introduce the problem setup of novel class discovery for point cloud segmentation and an overview of our method in Sec.3.1. We then describe our network architecture, including dual-level representation of point clouds in Sec.3.2. Finally, we present in detail our adaptive self-labeling framework for model learning that discovers the novel classes in Sec.3.3.

### 3.1 PROBLEM SETUP AND OVERVIEW

For the task of point cloud segmentation, the novel class discovery problem aims to learn to classify 3D points of a scene into known and novel semantic classes from a dataset consisting of annotated points from the known classes and unlabeled points from novel ones.

Formally, we consider a training set of 3D scenes, where each scene comprises two parts: 1) an annotated part of the scene $\{(x_n^s, y_n^s)\}_{n=1}^N$, which belongs to the known classes $C^s$ and consists of original point clouds along with the corresponding ground truth labels for each point; 2) a unknown part of the scene $\{(x_m^u)\}_{m=1}^M$, which belongs to the novel classes $C^u$ and does not contain any label information. These two sets $C^s$ and $C^u$ are mutually exclusive, i.e., $C^s \cap C^u = \emptyset$. Our goal is to learn a point cloud segmentation network that can accurately segment new scenes in a test set, each of which includes both known and novel classes.

To tackle the challenge of discovering novel classes in point clouds, we introduce a dual-level adaptive self-labeling framework to learn a segmentation network for both known and unknown classes. The key idea of our method includes two aspects: 1) utilizing the spatial smooth prior of point clouds to generate regions and developing a dual-level representation that incorporates regional consistency into the point-level classifier learning; 2) generating imbalance pseudo labels with a novel adaptive regularization. An overview of our framework is illustrated in Fig.1.

---

**Algorithm 1:** Semi-relaxed Optimal Transport Algorithm

---

**Function** `Self-Labeling`$(-\log \mathbf{P}, \gamma, \epsilon)$
   |    $\mathbf{K} = \exp(\log \mathbf{P}/\epsilon), \quad f \leftarrow \frac{\gamma}{\gamma+\epsilon}$
   |    $\boldsymbol{\mu}, \boldsymbol{\nu} \leftarrow \frac{1}{M}\mathbf{1}_M, \frac{1}{|C^u|}\mathbf{1}_{|C^u|}$ //Marginal distribution
   |    $\mathbf{b}_0 \leftarrow \mathbf{1}_{|C^u|}$ //Initialize $\mathbf{b}$
   |    **while** $\|\mathbf{b}^{t+1} - \mathbf{b}^t\| < 1e-4$ **do**
   |       |    $\mathbf{a} \leftarrow \frac{\boldsymbol{\mu}}{\mathbf{K}\mathbf{b}_t}$
   |       |    $\mathbf{b}_{t+1} \leftarrow (\frac{\boldsymbol{\nu}}{\mathbf{K}^\top \mathbf{a}})^f$
   |    **end**
   |    $\mathbf{Q} \leftarrow M\mathrm{diag}(\mathbf{a})\mathbf{K}\mathrm{diag}(\mathbf{b})$
   |    **return Q**
**end**

---

### 3.2 MODEL ARCHITECTURE

We adopt a generic segmentation model architecture consisting of a feature encoder for the input point cloud and a classifier head to generate the point-wise class label prediction. Note that in order to capture both known and novel classes, our feature encoder is shared by all the classes $C^s \cup C^u$ and the output space of our classifier head also includes known and novel classes. Below we first introduce our feature representation and encoder, followed by the classifier head.

**Dual-level Representation.** Instead of treating each point independently, we intend to exploit the spatial smoothness prior of 3D objects in our representation learning. To this end, we adopt an dual-level representation of point clouds that describes the input scene at different granularity. Specifically, given an input point cloud $\mathbf{X}$, we first use a backbone network $f_\theta$ to compute a point-wise feature $\mathbf{Z}^p = \{\mathbf{z}_i^p\}$, where $\mathbf{z}_i^p \in \mathbb{R}^{D \times 1}$. In this work, we employ MinkowskiUNet (Choy et al., 2019) for the backbone. In addition, we cluster points into regions based on their coordinates and then compute regional features by average pooling of point features. Concretely, during training, we first utilize DBSCAN (Ester et al., 1996) to generate $K_i$ regions, $\mathcal{R} = \{r_k\}_{k=1}^{K_i}$, for unlabeled point in sample $i$, and calculate the regional features as follows,

$$\{r_k\}_{k=1}^{K_i} \leftarrow \mathrm{DBSCAN}(\{x_i^u\}_{i=1}^M), \quad \mathbf{z}_k^r = \mathrm{AvgPool}\{\mathbf{z}_i^p | \mathbf{z}_i^p = f_\theta(x_i^u), \; x_i^u \in r_k\}, \quad (1)$$

where $\mathbf{z}_k^r$ is the feature of region $r_i$. Such a dual-level representation allows us to enforce regional consistency in representation learning.

**Prototype-based Classifier.** We adopt a prototype-based classifier design for generating the point-wise predictions. Specifically, we introduce a set of prototypes for known and novel classes, denoted as $h = [h^s, h^u] \in \mathbb{R}^{D \times (|C^s|+|C^u|)}$. For each point or region, we compute the cosine similarity between its feature and the prototypes, followed by Softmax to predict the class probabilities. Here we use the same set of prototypes for the points and regions, which enforces a consistency constraint within each region and results in a more compact representation for each class.

### 3.3 ADAPTIVE SELF-LABELING FRAMEWORK

To cope with inherently class-imbalanced data, we propose a novel adaptive self-labeling framework, which dynamically generates imbalanced pseudo-labels for model learning. To this end, we adopt the following loss function for the known and novel classes,

$$\mathcal{L} = \mathcal{L}_s + \alpha\mathcal{L}_u^p + \beta\mathcal{L}_u^r, \quad (2)$$

where $\mathcal{L}_s$ is the cross-entropy loss for known classes, $\mathcal{L}_u^p$ is point-level loss and $\mathcal{L}_u^r$ is region-level loss for novel classes. $\alpha$ and $\beta$ are weight parameters. For the novel classes, we first generate pseudo labels for points and regions by solving a semi-relaxed Optimal Transport problem and then adopt the cross-entropy loss with the generated labels. The pseudo-code of our algorithm is shown in Appendix B and below we will focus on our novel pseudo label generation process.

**Imbalanced Pseudo Label Generation.** The pseudo label generation for balanced classes can be formulated as an optimal transport problem as follows (Asano et al., 2020; Zhang et al., 2023a):

$$\min_{\mathbf{Q}} \frac{1}{M} \langle \mathbf{Q}, -\log \mathbf{P}^u \rangle_F, \quad \text{s.t.} \; \mathbf{Q}\mathbf{1}_{|C^u|} = \mathbf{1}_M, \mathbf{Q}^\top \mathbf{1}_M = \frac{M}{|C^u|}\mathbf{1}_{|C^u|}, \quad (3)$$

where $\mathbf{Q} \in \mathbb{R}^{M \times |C^u|}$ are the pseudo labels of unlabeled data, $<,>_F$ is Frobenius inner product and $\mathbf{P}^u$ are the output probabilities of the model. For imbalanced point cloud data, we relax the second constraint on the class sizes in Equ 3, which leads to a parameterized semi-relaxed optimal transport problem as below:

$$\min_{\mathbf{Q}} \mathcal{F}_u(\mathbf{Q}, \gamma) = \frac{1}{M} \langle \mathbf{Q}, -\log \mathbf{P}^u \rangle_F + \gamma KL(\frac{1}{M}\mathbf{Q}^\top \mathbf{1}_M, \frac{1}{|C^u|}\mathbf{1}_{|C^u|})$$

$$\text{s.t. } \mathbf{Q} \in \{\mathbf{Q} \in \mathbb{R}^{M \times |C^u|} | \mathbf{Q}\mathbf{1}_{|C^u|} = \mathbf{1}_M\}, \tag{4}$$

where $\gamma$ is a weight coefficient for balancing the constraint on cluster size distribution in the second term. We note that, add an entropy term $-\epsilon \mathcal{H}(\frac{1}{M}\mathbf{Q})$ to Equ.(4) and for any given $\gamma$, this entropic semi-relaxed OT problem can be efficiently solved by fast scaling algorithms (Cuturi, 2013; Chizat et al., 2018). Alg.1 outlines the optimization process, and further details are provided in Appendix A.

In this work, we propose a novel adaptive regularization strategy that adjusts the weight $\gamma$ according to the progress of model learning. This enables more flexible pseudo-label generation, which significantly improves the representation learning of novel classes. The details of our adaptation strategy will be illustrated subsequently.

**Adaptive Regularization Strategy.** The objective in Equ.(4) aims to strike a balance between the distribution represented by model prediction $\mathbf{P}^u$ and the uniform prior distribution. A large $\gamma$ tends to prevent the model from learning a degenerate solution, e.g. assigning all the samples into a single novel class, but it also restrict the model's capacity in learning the imbalanced data. One of our key insight is that the imbalanced NCD learning requires an adaptive strategy for setting the value of $\gamma$ during the training. Intuitively, in the early training stage where the model performance is relatively poor, a larger constraint on $\mathbf{Q}^\top \mathbf{1}_M$ is needed to prevent degenerate solutions. As the training progresses, the model gradually learns meaningful clusters for novel classes, and the constraint should be relaxed in order to increase the flexibility of pseudo label generation.

To achieve that, we develop an annealing-like strategy for adjusting $\gamma$, which is inspired by the adaptive learning rate method, ReduceLROnPlateau, that reduces the learning rate when the loss does not decrease. Here we employ the KL term in Equ.(4) as a guide for decreasing $\gamma$, as the value of the KL term reflects the relationship between the distribution of pseudo labels and the uniform distribution. Specifically, our formulation for the adaptive regularization factor is as follows:

$$\gamma_{t+1} = \lambda \gamma_t \quad \text{if } KL(\frac{1}{M}\mathbf{Q}^\top \mathbf{1}_M, \frac{1}{|C^u|}\mathbf{1}_{|C^u|}) \leq \rho \text{ consecutively for } T \text{ iterations,} \tag{5}$$

where $\rho, \lambda, T$ and $\gamma_0$ are hyperparameters. Compared to typical step decay and cosine decay strategies, our adaptive strategy is aware of the model learning process and allows for more flexible control of $\gamma$ based on the characteristics of the input itself.

*Hyperparameter Search.* To search the values of our hyperparameters, we design an indicator score that can be computed on the training dataset. Specifically, our indicator regularizes the total loss in Equ 2 with a KL term that measures the distance between the distribution of novel classes and the uniform distribution. Formally, the indicator is defined as follows:

$$\mathcal{I} = \mathcal{L} + \gamma KL(\frac{1}{M}\mathbf{Q}^\top \mathbf{1}_M, \frac{1}{|C^u|}\mathbf{1}_{|C^u|}), \tag{6}$$

where $\gamma$ is obtained by Equ.(5). Empirically, this indicator score provides a balanced evaluation of the model's performance on the known and novel classes.

## 4 EXPERIMENTS

### 4.1 EXPERIMENTAL SETUP

**Dataset.** We perform evaluation of our method on the widely-used SemanticKITTI (Geiger et al., 2012; Behley et al., 2021; 2019) and SemanticPOSS (Pan et al., 2020) datasets. The SemanticKITTI dataset consists of 19 semantic classes, while the SemanticPOSS dataset contains 13 semantic classes. Both datasets suffer from intrinsic class imbalances. For fair comparison with existing works (Riz et al., 2023), we divide the dataset into 4 splits and select one split as novel classes, while treating the remain splits as the known classes. Additionally, to assess the effectiveness of our method under more challenging conditions, we further split the SemanticPOSS dataset into two parts, selecting one part as novel classes. The dataset details are provided in Appendix C.

Table 1: The novel class discovery results on SemanticPOSS dataset. 'Number' denotes the number of points. 'Full' denotes the results obtained by supervised learning. The gray values are the novel classes in each split.

| Split | Method | bike | build. | car | cone. | fence | grou. | pers. | plants | pole | rider | traf. | trashc. | trunk | Novel | Known | All |
|---|---|---|---|---|---|---|---|---|---|---|---|---|---|---|---|---|---|
| | Full | 45.0 | 83.3 | 52.0 | 36.5 | 46.7 | 77.6 | 68.2 | 77.7 | 36.0 | 58.9 | 30.3 | 4.2 | 14.4 | - | - | 48.5 |
| 0 | EUMS | 25.7 | 4.0 | 0.6 | 16.4 | 29.4 | 36.8 | 43.8 | 28.5 | 13.1 | 26.8 | 18.2 | 3.3 | 16.9 | 17.4 | 21.5 | 20.3 |
| | NOPS | 35.5 | 30.4 | 1.2 | 13.5 | 24.1 | 69.1 | 44.7 | 42.1 | 19.2 | 47.7 | 24.4 | 8.2 | 21.8 | 35.7 | 26.6 | 29.4 |
| | Ours | 46.3 | **51.5** | **6.0** | 35.7 | 48.5 | **83.0** | 67.9 | **53.1** | 35.5 | 59.3 | 31.0 | 2.8 | 15.5 | **48.4** | 38.0 | 41.2 |
| 1 | EUMS | 15.2 | 68.0 | 28.0 | 24.0 | 11.9 | 75.1 | 36.0 | 74.5 | 26.9 | 48.6 | 26.0 | 5.6 | 23.1 | 21.0 | 40.0 | 35.6 |
| | NOPS | 29.4 | 71.4 | 28.7 | 12.2 | 3.9 | 78.2 | **56.8** | 74.2 | 18.3 | 38.9 | 23.3 | 13.7 | 23.5 | 30.0 | 38.2 | 36.4 |
| | Ours | **31.5** | 83.2 | 48.7 | 25.4 | **23.9** | 77.3 | 53.1 | 77.1 | 32.5 | 57.3 | 35.0 | 9.3 | 18.0 | **36.2** | 46.4 | 44.0 |
| 2 | EUMS | 40.1 | 69.5 | 27.7 | 13.5 | 34.9 | 76.0 | 54.7 | 75.6 | 5.3 | 39.2 | 7.8 | 8.5 | 11.9 | 8.3 | 44.0 | 35.7 |
| | NOPS | 37.2 | 71.8 | 29.7 | 14.6 | 28.4 | 77.5 | 52.1 | 73.0 | **11.5** | 47.1 | 0.5 | 10.2 | **14.8** | 9.0 | 44.2 | 36.0 |
| | Ours | 45.3 | 82.8 | 49.8 | 28.4 | 46.3 | 76.7 | 66.2 | 77.2 | 10.9 | 58.4 | **18.6** | 7.3 | 8.2 | **12.6** | 53.8 | 44.3 |
| 3 | EUMS | 41.2 | 70.7 | 28.1 | **4.3** | 38.3 | 76.7 | 38.3 | 75.4 | 25.8 | 34.3 | 28.3 | 0.4 | 24.4 | 13.0 | 44.7 | 37.4 |
| | NOPS | 38.6 | 70.4 | 30.9 | 0.0 | 29.4 | 76.5 | 56.0 | 71.8 | 17.0 | 31.9 | 26.2 | 1.0 | 22.6 | 10.9 | 43.9 | 36.3 |
| | Ours | 45.5 | 82.9 | 47.7 | 0.0 | 45.1 | 77.8 | 66.3 | 77.7 | 34.3 | **49.1** | 35.6 | **4.0** | 15.3 | **17.7** | 52.8 | 44.7 |

Table 2: A more harder splits on SemanticPOSS dataset. The gray values are the novel classes in each split. We reimplement NOPS based on their released code.

| Split | Method | bike | build. | car | cone. | fence | grou. | pers. | plants | pole | rider | traf. | trashc. | trunk | Novel | Known | All |
|---|---|---|---|---|---|---|---|---|---|---|---|---|---|---|---|---|---|
| | Full | 45.0 | 83.3 | 52.0 | 36.5 | 46.7 | 77.6 | 68.2 | 77.7 | 36.0 | 58.9 | 30.3 | 4.2 | 14.4 | - | - | 48.5 |
| 0 | NOPS | 37.4 | 22.9 | 8.1 | 0.0 | 30.3 | 78.9 | 4.8 | 72.9 | **1.0** | 42.9 | 25.8 | 9.2 | **9.6** | 7.7 | 42.5 | 26.5 |
| | Ours | 46.0 | **26.1** | **27.5** | **2.8** | 46.9 | 77.6 | **35.0** | 77.8 | 0.2 | 58.6 | 30.5 | 3.2 | 0.0 | **15.3** | 48.7 | 33.2 |
| 1 | NOPS | 6.1 | 71.3 | 35.6 | 21.2 | 3.1 | **42.9** | 44.5 | 26.0 | 24.4 | **0.7** | 0.6 | **0.1** | 24.8 | 11.4 | 37.0 | 23.2 |
| | Ours | **26.3** | 82.0 | 51.4 | 18.0 | **10.4** | 40.0 | 67.5 | **32.5** | 31.2 | 0.0 | **6.3** | 0.0 | 11.7 | **16.5** | 44.5 | 29.4 |

**Evaluation Metric.** Following the official benchmark guidelines in SemanticKITTI and SemanticPOSS, we evaluate on sequence 08 and 03, respectively. These sequences contain both known and novel classes. For the known classes, we report the Intersection over Union (IoU) for each class. Regarding the novel classes, we employ the Hungarian algorithm to initially match cluster labels with their corresponding ground truth labels. Subsequently, we present the IoU values for each of these novel classes. Additionally, we calculate the mean IoU across all known and novel classes.

**Implementation Details.** We follow (Riz et al., 2023) to adopt the MinkowskiUNet-34C (Choy et al., 2019) network as our backbone. For the input point cloud, we set the voxel size as 0.05 and utilize the scale and rotation augmentation to generate two views. The scale range is from 0.95 to 1.05, and the rotation range is from $-\pi/20$ to $\pi/20$ for three axes. We train 10 epochs and set batch size as 4 for all experiments. The optimizer is Adamw, and the initial learning rate is 1e-3, which will decrease to 1e-5 by a cosine schedule. For the hyperparameters, we set $\alpha = \beta = 1$, fix $\lambda$ at 0.5 and analyze DBSCAN in Appendix K. We choose $T = 10$ and $\rho = 0.005$ based on the indicator mentioned in Sec.3.3 and analyze them in the ablation study. Both the point- and region-level self-labeling algorithms employ the same parameters. All experiments are conducted on a single NVIDIA A100.

## 4.2 RESULTS

**SemanticPOSS Dataset.** As presented in Tab.1, our approach exhibits significant improvements on novel over the previous method across all four splits. Specifically, we achieve an increase of **12.7%** and **6.2%** in split 0 and 1, respectively. It is worth noting that the fully supervised upper bounds for novel classes in split 0 and 1 are 72.7% and 53.3%, respectively, surpassing our method by 24.3% and 17.1%. In the more challenging split 2 and split 3, we observe gains of **3.6%** and **4.7%**, respectively. The corresponding upper bounds for these splits are 26.9% and 33.2%, indicating their increased difficulty compared to splits 0 and 1. Compared with NOPS, our method achieves improvements in head(7.5%), medium(8.9%), and tail(6.8%) classes. On average, we achieve an IoU of 30.2% for novel classes across all four splits, outperforming NOPS (22.5%) by **7.7%**. Details of dataset distribution are in Appendix C. Note that the improvement in known classes can be

Table 3: The novel class discovery results on the SemanticKITTI dataset. 'Full' denotes the results obtained by supervised learning. The four groups represent the four splits in turn, and the gray values are the novel classes in each split.

| Method | bi.cle | b.clst | build. | car | fence | mt.cle | m.clst | oth-g. | oth-v. | park. | pers. | pole | road | side2. | terra. | traff. | truck | trunk | veget. | Novel | Known | All |
|---|---|---|---|---|---|---|---|---|---|---|---|---|---|---|---|---|---|---|---|---|---|---|
| Full | 2.9 | 55.4 | 89.5 | 93.5 | 27.9 | 27.4 | 0.0 | 0.9 | 19.9 | 35.8 | 31.2 | 60.0 | 93.5 | 77.8 | 62.0 | 39.8 | 50.8 | 53.9 | 87.0 | - | - | 47.9 |
| EUMS | 5.3 | 40.0 | 15.8 | 79.2 | 9.0 | 16.9 | 2.5 | 0.1 | 11.4 | 14.4 | 12.7 | 29.2 | 42.6 | 26.1 | 0.1 | 10.3 | 47.4 | 37.9 | 38.4 | 24.6 | 21.1 | 23.1 |
| NOPS | 5.6 | 47.8 | 52.7 | 82.6 | 13.8 | 25.6 | 1.4 | 1.7 | 14.5 | 19.8 | 25.9 | 32.1 | 56.7 | 8.1 | 23.8 | 14.3 | 49.4 | 36.2 | 44.2 | 37.1 | 26.5 | 29.3 |
| Ours | 5.5 | 51.1 | 74.6 | 92.3 | 29.8 | 22.8 | 0.0 | 0.0 | 23.3 | 24.8 | 27.7 | 59.7 | 41.4 | 22.5 | 23.6 | 39.3 | 43.6 | 51.1 | 66.4 | 45.7 | 33.7 | 36.8 |
| EUMS | 7.5 | 42.4 | 80.0 | 76.8 | 8.6 | 19.6 | 1.4 | 0.6 | 12.0 | 14.1 | 14.0 | 40.7 | 86.3 | 66.5 | 56.3 | 12.0 | 44.8 | 20.9 | 72.4 | 24.2 | 37.1 | 35.6 |
| NOPS | 7.4 | 51.2 | 84.5 | 50.9 | 7.3 | 28.9 | 1.8 | 0.0 | 22.2 | 19.4 | 30.4 | 37.6 | 90.1 | 72.2 | 60.8 | 16.8 | 57.3 | 49.3 | 85.1 | 25.4 | 46.2 | 40.7 |
| Ours | 3.7 | 57.4 | 89.2 | 56.5 | 17.3 | 20.3 | 0.0 | 0.0 | 20.0 | 30.6 | 34.8 | 60.6 | 93.2 | 77.6 | 62.0 | 38.7 | 56.9 | 39.2 | 86.7 | 28.7 | 50.1 | 44.5 |
| EUMS | 8.3 | 50.8 | 83.0 | 88.1 | 17.9 | 2.8 | 2.3 | 0.2 | 3.2 | 25.4 | 25.0 | 20.2 | 88.3 | 71.0 | 57.9 | 8.6 | 27.2 | 38.4 | 77.0 | 12.4 | 42.2 | 36.6 |
| NOPS | 6.7 | 49.2 | 86.4 | 90.8 | 23.7 | 2.7 | 0.6 | 1.9 | 15.5 | 29.5 | 27.9 | 36.4 | 90.3 | 73.4 | 61.2 | 17.8 | 10.3 | 46.2 | 84.3 | 16.5 | 48.0 | 39.7 |
| Ours | 3.6 | 54.2 | 88.9 | 93.3 | 28.4 | 10.2 | 0.0 | 0.9 | 9.6 | 33.4 | 32.2 | 36.1 | 92.7 | 77.4 | 62.2 | 10.7 | 34.2 | 51.7 | 86.9 | 20.1 | 50.4 | 42.5 |
| EUMS | 4.0 | 2.5 | 80.1 | 87.2 | 16.8 | 14.0 | 15.0 | 0.3 | 14.1 | 20.8 | 6.8 | 37.6 | 86.8 | 66.5 | 55.3 | 16.2 | 40.6 | 38.4 | 76.2 | 7.1 | 43.4 | 35.7 |
| NOPS | 2.3 | 27.8 | 86.0 | 89.9 | 23.1 | 24.5 | 2.9 | 3.1 | 18.2 | 30.1 | 16.3 | 39.9 | 90.7 | 73.5 | 61.0 | 17.4 | 49.8 | 44.0 | 83.2 | 12.4 | 49.0 | 41.2 |
| Ours | 2.6 | 32.5 | 88.7 | 93.3 | 28.1 | 24 | 0.1 | 1.0 | 23.7 | 35.6 | 15.3 | 59.8 | 93.2 | 77.6 | 61.4 | 37.8 | 56.6 | 52.1 | 86.7 | 12.6 | 54.6 | 45.8 |

attributed to the training of NOPS not converging. In the Appendix E, we compare our method and NOPS*, which training is converging. The results show that, compared with NOPS, NOPS* achieves sizeable improvements in known classes but drops a lot in novel classes. Therefore, our method still outperforms NOPS* by a sizeable margin.

To further verify that our method can alleviate the imbalanced problem, we divided the SemanticPOSS dataset into two splits, creating a more severe imbalance scenario that poses a greater challenge for clustering novel classes. As shown in Tab.2, on novel classes, our method outperforms NOPS significantly on both splits, with a margin of **7.6%** on split 0 and **5.0%** on split 1. Furthermore, for novel classes, we observe that our improvement mainly stems from the medium classes, such as person and bike. It's worth noting that NOPS employs extra training techniques, like multihead and overclustering, whereas we do not employ these techniques, further demonstrating our effectiveness. Additionally, we have created a video to visualize the results of NOPS and our method on Split 0, which is included in the supplementary material.

Moreover, to validate the effectiveness of our method in realistic scenarios, where the class number of unlabeled data is unknown, we estimate the number of novel classes by the method proposed by (Vaze et al., 2022), and conduct experiments. The results show our method still outperforms NOPS by a large margin. More details and analysis are in Appendix F.

**SemanticKITTI Dataset.** The results in Tab.3 demonstrate our superior performance compared to previous methods on different splits. Specifically, we achieve significant improvements of **8.6%**, **3.3%**, and **3.6%** on splits 0, 1, and 2, respectively, for novel classes. The supervised upper bounds for these splits are 82.0%, 42.4%, and 39.6%, respectively. In split 3, our results are slightly higher than NOPS by 0.2%, possibly due to the scarce presence of these novel classes in split 3. On average across all four splits, our approach achieves an IoU of 27.5%, surpassing NOPS (23.4%) by **4.1%** on novel classes. Please refer to Appendix D for detailed analysis and Appendix L for visualization.

### 4.3 ABLATION STUDY

**Component Analysis.** To analyze the effectiveness of each component, we conduct extensive experiments on split 0 of SemanticPOSS dataset. As shown in Tab.4, compared to baseline which employs equal-size constraints, imbalanced self-labelling improves performance by **4.2%**. The confusion matrix in Tab.2 indicates that except for the easily classifiable "ground", there is a significant improvement in the head and medium classes. This phenomenon is clearly depicted in Fig.3, where the predictions of the baseline exhibit noticeable noise. From the second and third rows of Tab.4, the adaptive regularization leads to a significant improvement of **8.2%** in split0 and **4.5%** in overall splits. As shown in Fig.2, adaptive regularization enhances the quality of pseudo-labels for each class, especially for the tail class (car) and head class (plants). We also the visualization the class distribution of pseudo label in Appendix G, which shows adaptive regularization provides greater flexibility than a fixed regularization factor According to the third and last rows of Tab.4, the inclusion

Table 4: Ablation study on SemanticPOSS. The results are on novel classes. ISL and AR denote imbalanced self-labelling and regularization respectively. Region denotes region-level learning. The last two columns respectively represent the average mIoU for split 0 and across all splits.

| ISL | AR | Region | Building | Car | Split0 Ground | Plants | Avg | Overall Avg |
|-----|-----|--------|----------|-----|--------|--------|-----|-------------|
| | | | 21.6 | 2.7 | 76.6 | 26.1 | 31.8 | 20.9 |
| ✓ | | | 27.6 | 3.1 | 81.2 | 32.1 | 36.0 | 23.9 |
| ✓ | ✓ | | 53.1 | 5.3 | 81.1 | 37.4 | 44.2 | 28.4 |
| ✓ | ✓ | ✓ | 51.5 | 6.0 | 83.0 | 53.1 | 48.4 | 30.2 |

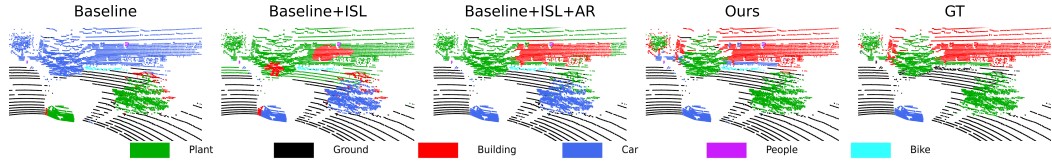

Figure 2: Confusion Matrix, GT on y-axis, Pseudo Label on x-axis. $(i, j)$ represents the % of GT in class $j$ assigned pseudo label $i$. We categorize 'plants' and 'ground' as head classes, 'building' as medium, and 'car' as tail classes.

Figure 3: Visualization analysis of different components.

Table 5: Analysis of adaptive regularization on SemanticPOSS dataset. GT denotes we directly assign the ground truth distribution of cluster size.

| $\gamma$ | 0.05 | 0.01 | 0.1 | 0.5 | 1 | 5 | $+\infty$ | GT | Adaptive |
|----------|------|------|-----|-----|---|---|-----------|-----|----------|
| Split0 | 10.1 | 33.8 | 33.3 | 36.0 | 32.2 | 33.8 | 31.8 | 32.5 | 44.2 |

of the region-level branch leads to a **4.2%** improvement. In Fig.2, there's a significant improvement in pseudo-labels for each category, particularly for the tail class (car) and the head class (plants). From Fig.3, it's evident that the region-level branch can correct cases where a single object is mistakenly labelled as multiple categories. Due to the utilization of spatial priors, where closely located coordinates are highly likely to belong to the same category, our region-level branch can correct misclassifications by considering neighbouring points, preventing splitting a single object into multiple entities. Those experiments validate the effectiveness of each component in our method.

**Adaptive Regularization and Hyperparameters Selection.** To analyze the impact of adaptive regularization, we compare it with various fixed regularization factors, as illustrated in Tab.5. We notice that employing a very small fixed $\gamma$, such as 0.05 as indicated in the table, results in a weak prior constraint, and the model tends to learn a degenerate solution where all samples are assigned to a single cluster. When the $\gamma$ increases to 0.5, the model achieves optimal results, but the increment decreases when the $\gamma$ further increases. Compared with adaptive $\gamma$, the optimal results of fixed $\gamma$ is nearly 8.2% lower, demonstrating that the adoption of an adaptive $\gamma$ not only enhances the model's flexibility but also prevents any performance degradation. Furthermore, we experiment with adopting the GT class distribution and substituting the KL constraint in Equ.(4) with an equality constraint. Surprisingly, the results indicate that the GT class distribution constraint is not the optimal solution for addressing imbalanced novel class discovery. We visualize the $\gamma$ curves for SemanticPOSS in four splits, as shown in Fig.4. Split 0 exhibits the highest rate of change, followed by Split 1, while Splits 2 and 3 remain constant, indicating that our strategy is adaptive to each dataset.

To further validate the effectiveness of adjusting $\gamma$ based on KL divergence, we also compare it with typical step decay and cosine annealing strategies. For the step decay, we set the initial $\gamma$ to 1 and

Table 6: Step decay results

| $\lambda$ | 0.1 | 0.3 | 0.5 | 0.7 | 0.9 |
|---|---|---|---|---|---|
| Step decay | 34.0 | 34.2 | 32.9 | 34.6 | 33.3 |

Table 7: Cosine annealing results

| min $\gamma$ | 0.1 | 0.05 | 0.01 | 0.005 | 0.001 |
|---|---|---|---|---|---|
| Cosine annealing | 32.2 | 32.5 | 35.8 | 36.1 | 32.0 |

Table 8: The results for different values of $\rho$

| $\rho$ | 0.05 | 0.01 | 0.005 | 0.001 |
|---|---|---|---|---|
| Split0 | 10.08 | 45.84 | 44.21 | 33.11 |

Table 9: The results for different values of $T$

| $T$ | 5 | 10 | 20 | 30 |
|---|---|---|---|---|
| Split0 | 44.46 | 44.21 | 44.12 | 32.48 |

Figure 4: $\gamma$ variation

Figure 5: Selecting $\rho$

Figure 6: Selecting $T$

Table 10: Ablation the alternate design of region-level prototype on split 0 of SemanticPOSS dataset. The results are on novel classes.

| Prototype Sharing | building | car | ground | plants | avg |
|---|---|---|---|---|---|
| $\times$ | 25.4 | 9.5 | 81.6 | 31.0 | 36.9 |
| $\checkmark$ | 51.5 | 6.0 | 83.0 | 53.1 | 48.4 |

decay it by multiplying it with $\lambda$ every epoch. For the cosine annealing approach, we also set the initial $\gamma$ to 1 and reduce it to the minimum value (min $\gamma$). From the Tab.6 and Tab.7, we observe that the results of simple step decay and cosine annealing are nearly 10% worse than adaptive $\gamma$ (which is 44.2). We believe that this two typical strategies lack flexibility compared to adaptive $\gamma$. They might not facilitate the adaptive control of the $\gamma$ decay process based on the model learning process.

To choose the hyperparameters $\rho$ and $T$ according to the indicator outlined in Sec.3.3, we conduct experiments for various values of $\rho$ and $T$. The results are displayed in Tab.8 and Tab.9. Additionally, we plot the indicator's curve for each experiment in Fig.5 and 6. The plots reveal that when $\rho$ falls within the range of 0.01 to 0.005, and $T$ is set between 5 and 20, the indicator value remains low while achieving a high novel IoU. Those results demonstrate the efficiency of our hyperparameters selection strategy and the robustness of our method.

**Prototype sharing of Region-level Learning.** We conduct experiments without sharing prototypes, and the results are depicted in Tab.10. It is noteworthy that utilizing two isolated prototypes results in a significant drop of nearly 10% in performance in the novel class. The analysis of similarity between point and region prototypes in Appendix H.2 further illustrates that having two isolated prototypes leads to disparities in point-wise and region-wise learning directions. Conversely, sharing prototypes promotes the model to learn more compact and coherent representations.

## 5 CONCLUSION

In this paper, we propose a novel dual-level adaptive self-labeling framework for novel class discovery in point cloud segmentation. Our framework formulates the pseudo label generation process as a Semi-relaxed Optimal Transport problem and incorporate a novel data-dependent adaptive regularization factor to decay the constraint of the uniform prior based on the distribution of pseudo label, thereby enable more flexibility to generate higher-quality imbalanced pseudo labels for model learning. In addition, we develop a dual-level representation that leverages the spatial prior to generate region representation, which reduce the noise in generated segmentation and enhance point-level classifier learning. Furthermore, we propose a hyperparameters search strategy based on training set. Extensive experiments on two widely-used datasets, SemanticKITTI and SemanticPOSS demonstrate the effectiveness of each component and the superiority of our method.

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

# A  SEMI-RELAXED OPTIMAL TRANSPORT

In this section, we first detail the advantage of our formulation and then show how to solve the semi-relaxed optimal transport by an efficient scaling algorithm, finally analyze the difference of our algorithm with Zhang et al. (2023a) in detail. Without losing generality, we consider the generic formulation as follows:

$$\min_{\mathbf{Q}} \langle \mathbf{Q}, \mathbf{C} \rangle_F + \gamma KL(\mathbf{Q}^\top \mathbf{1}_M, \boldsymbol{\mu}) \tag{7}$$

$$\text{s.t. } \mathbf{Q} \in \{\mathbf{Q} \in \mathbb{R}^{M \times N} | \mathbf{Q}\mathbf{1}_N = \boldsymbol{\nu}\}, \tag{8}$$

where $\boldsymbol{\mu}, \boldsymbol{\nu}$ are the prior marginal distribution of $\mathbf{Q}$, and $\mathbf{C}$ is the cost matrix.

As analyzed in Appendix H.1, the representation and prototype will gradually bootstrap. Therefore, the imbalanced property of dataset will be reflected in the model's prediction $P$. OT generate pseudo labels ($Q$) based on P and several distribution constraints. Unlike typical OT, which imposes two equality constraints and enforces a uniform distribution, our relaxed OT utilizes a relaxed KL constraint on cluster size. In the optimization of our relaxed OT, the optimal $Q$ for the $< Q, -\log P >$ is to assign $Q$ based on the largest prediction in $P$, thus capturing the imbalanced property of $P$, and the KL constraints enforce the marginal distribution of $Q$ not far away from the prior uniform distribution, avoiding degenerate solution while providing some flexibility. Consequently, the optimal $Q$ in our relaxed OT should take into account both the inherent imbalanced distribution of classes in $P$ and a prior uniform distribution, thus generating pseudo labels reflecting imbalanced characteristics of $P$.

However, the above formulation is well-suited with our problem, the time complexity of this algorithm is quadratic, which is unaffordable for large-scale problems. To solve it efficiently, motivated by Cuturi (2013), we first introduce an entropic constraints, $-\epsilon \mathcal{H}(\mathbf{Q})$. Due to

$$\langle \mathbf{Q}, \mathbf{C} \rangle_F - \epsilon \mathcal{H}(\mathbf{Q}) = \epsilon \langle \mathbf{Q}, \mathbf{C}/\epsilon + \log \mathbf{Q} \rangle_F = \epsilon \langle \mathbf{Q}, \log \frac{\mathbf{Q}}{e^{-\mathbf{C}/\epsilon}} \rangle_F, \tag{9}$$

The entropic semi-relaxed optimal transport can be reformulated as:

$$\epsilon \langle \mathbf{Q}, \log \frac{\mathbf{Q}}{e^{-\mathbf{C}/\epsilon}} \rangle_F + \gamma KL(\mathbf{Q}^\top \mathbf{1}_M, \boldsymbol{\mu}) \tag{10}$$

$$\text{s.t. } \mathbf{Q} \in \{\mathbf{Q} \in \mathbb{R}^{M \times N} | \mathbf{Q}\mathbf{1}_N = \boldsymbol{\nu}\}. \tag{11}$$

Then, this problem can be approximately solved by an efficient scaling algorithm. Refer to Chizat et al. (2018) for more derivation.

Zhang et al. (2023a) introduce an imbalanced self-labeling learning framework that tackles the issue of novel class discovery in the long-tailed scenario. To generate imbalanced pseudo label they introduce an auxiliary variable $\mathbf{w} \in \mathbb{R}^N$, which is dynamically inferred during learning and encodes a proper constraint on the cluster-size distribution. Their formulation can be written:

$$\min_{\mathbf{Q}} \langle \mathbf{Q}, \mathbf{C} \rangle_F + \gamma KL(\mathbf{w}, \boldsymbol{\mu}) \tag{12}$$

$$\text{s.t. } \mathbf{Q} \in \{\mathbf{Q} \in \mathbb{R}^{M \times N} | \mathbf{Q}\mathbf{1}_N = \boldsymbol{\nu}, \mathbf{Q}^\top \mathbf{1}_M = \mathbf{w}\}, \tag{13}$$

Unlike us, they adopt a fixed $\gamma$ in their approach. To optim Equ.(12), they propose a bi-level optimization strategy, which alternately estimates cluster distributions and generates pseudo labels by solving an optimal transport problem. Specifically, they start from a fixed $\mathbf{w}$ and first minimize Equ.(12) $w.r.t$ $\mathbf{Q}$. As the KL constraint term remains constant, the task turns into a standard optimal transport problem, which can be efficiently solved by the Sinkhorn-Knopp Algorithm. Then they optimize Equ.(12) $w.r.t$ $\mathbf{w}$ with simple gradient descent.

While their bi-level optimization approximates the objective function, it consumes a significant amount of time compared to direct application of the light-speed scaling algorithm. Additionally, it introduces additional hyperparameters $\mathbf{w}$ for inner-loop optimization. As shown in Fig. 7, compared to bi-level optimization strategy proposed by Zhang et al. (2023a), the scaling algorithm is more fast, making it possible to solve large scale problem.

# B  PSEUDO CODE OF OUR METHOD

To provide a clearer description of our method, we detail our method in Alg.2.

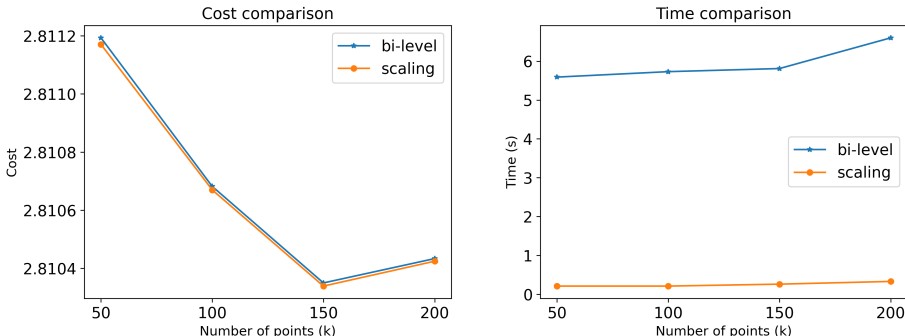

Figure 7: Comparison of two optimization algorithm.

---

**Algorithm 2:** Dual-level Imbalanced-aware Self-labeling and Learning Algorithm

---

**Input:** $\mathcal{D}^{tr} = \{\mathbf{X}^s, \mathbf{X}^u\}$, softmax function $\sigma$, encoder $f_\theta$, classifier $h = [h^s, h^u] \in \mathbb{R}^{D \times (|C^s| + |C^u|)}$,
  $\boldsymbol{\mu} = \mathbf{1}$, uniform distribution $\boldsymbol{\nu}$, hyperparameters $\gamma_p, \gamma_r, \lambda, \rho, T, \alpha, \beta, \epsilon$

**for** $e \in 1, 2, .., Epoch$ **do**
  **for** $s \in 1, 2, ..., Step$ **do**
    $\{(x_i^s, y_i^s)\}_{i=0}^N, \{x_j^u\}_{j=0}^M \leftarrow \text{Sample}(\mathcal{D}^{tr})$
    //Cluster point into different regions by DBSCAN
    $\{r_k\}_{k=0}^K \leftarrow DBSCAN(\{x_j^u\}_{j=0}^M)$
    //Forward the model
    $p^s = \text{softmax}(h^s \circ f_\theta(x^s)/\tau)$
    $\mathbf{z}_p = f_\theta(x^u), \mathbf{z}_r = \{\text{AvgPool}(\mathbf{z}_p) | r_k \text{ is same}\}$
    $p_p^u = \text{softmax}(h^u \circ \mathbf{z}_p/\tau), p_r^u = \text{softmax}(h^u \circ \mathbf{z}_r/\tau)$
    //CE loss for known classes
    $\mathcal{L}_s = -\frac{1}{N} \sum_{i=1}^N y_i^s \log p_i^s$
    //Point level self-labeling
    $\mathbf{Q}_p^u = \text{Self-Labeling}(-\log \mathbf{P}_p^u, \gamma_t^p, \epsilon)$
    $\mathcal{L}_p^u = \frac{1}{M} \langle \mathbf{Q}_p^u, -\log \mathbf{P}_r^u \rangle_F$
    //Region level self-labeling
    $\mathbf{Q}_r^u = \text{Self-Labeling}(-\log \mathbf{P}_r^u, \gamma_t^r, \epsilon)$
    $\mathcal{L}_u^r = \frac{1}{K} \langle \mathbf{Q}_r^u, -\log \mathbf{P}_r^u \rangle_F$
    //Update $\gamma_{t+1}^p$ and $\gamma_{t+1}^r$
    **if** $KL(\frac{1}{M}\mathbf{Q}_p^{u\top}\mathbf{1}_M, \boldsymbol{\nu}) \leq \rho$ *consistently for T iterations* **then**
      $\mid \quad \gamma_{i+1}^p = \lambda\gamma_i^p$
    **end**
    **if** $KL(\frac{1}{K}\mathbf{Q}_r^{u\top}\mathbf{1}_M, \boldsymbol{\nu}) \leq \rho$ *consistently for T iterations* **then**
      $\mid \quad \gamma_{i+1}^r = \lambda\gamma_i^r$
    **end**
    //Total loss
    minimize $\mathcal{L}_s + \alpha\mathcal{L}_u^p + \beta\mathcal{L}_u^r$ *w.r.t* $\theta$
  **end**
**end**

---

## C  DATASET SPLITS

We follow Riz et al. (2023) and divide SemanticKITTI and SemanticPOSS into four splits, as shown in Tab.11 and 12. It is worth noting that, to avoid the most frequent novel class affecting the other classes and to better utilize the semantic relationships between known and novel classes, they balance the distribution of the novel classes in each split. Fig.8's left and middle plots illustrate the distribution of SemanticKITTI and SemanticPOSS in each split. However, deliberately selecting novel classes in this manner is not reasonable as it avoids addressing the issue of point cloud data imbalance. To validate the generalization of our algorithm, we conduct experiments on a more challenging benchmark. In this benchmark, we cross-select half of the classes from SemanticPOSS as novel classes, as shown in the right side of Fig.8. We analyze the results in Sec.4.2.

Table 11: The detail of novel classes in each split.

| Split | SemanticKITTI | SemanticPOSS |
|---|---|---|
| 0 | building,road,sidewalk,terrain,vegetation | building,car,ground,plants |
| 1 | car,fence,other-ground,parking,trunk | bike,fence,person |
| 2 | motorcycle,other-vehicle,pole,traffic-sign,truck | pole,traffic-sign,trunk |
| 3 | bicycle,bicyclist,motorcyclist,person | cone-stone,rider,trashcan |

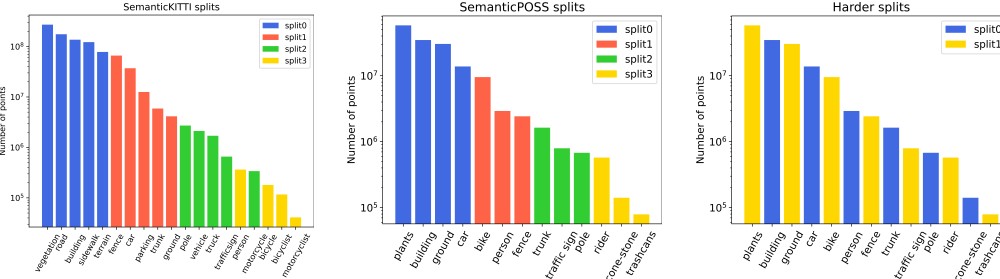

Figure 8: Distribution plot of the SemanticPOSS dataset. Each class has been assigned the color of the split in which it has to be considered novel.

Table 12: The detail of novel classes on challenging setting of SemanticPOSS dataset.

| Split | SemanticPOSS |
|---|---|
| 0 | building,car,ground,plants,bike,fence,person |
| 1 | pole,traffic-sign,trunk,cone-stone,rider,trashcan |

## D  MORE DETAIL ANALYSIS

To validate that our method can handle imbalanced data distributions, we compute the mIoU for head classes, medium classes, and tail classes within each split. The results for both datasets are shown in Tab.13 and Tab.14. Specifically, for SemanticPOSS, in the first split, there are a total of 4 novel classes. We choose the two classes with the highest count as head classes, and the remaining two are designated as medium and tail classes. In the other splits, which all have 3 novel classes, we sort them by size and assign them as head, medium, and tail classes accordingly. For SemanticKITTI, we designated the largest class and the smallest class within each split as head and tail classes, respectively, with the rest categorized as medium classes.

As shown in Tab.13, in SemanticPOSS, our approach achieve improvements of 7.5%, 8.9%, and 6.8% for head, medium, and tail classes, respectively, compared to NOPS. Similarly, in SemanticKITTI, our method yielded improvements of 7.7%, 3.9%, and 1.1% for head, medium, and tail classes, respectively. This indicates that our method provides improvements for head, medium, and tail classes, demonstrating its effectiveness in addressing imbalanced data scenarios. It's worth noting that NOPS utilizes additional techniques such as multihead and overclustering during training to enhance its performance, whereas we do not employ these techniques, further demonstrating the effectiveness of our method.

## E  IMPROVED NOPS

We observe that NOPS employs a learning rate of 0.01 in conjunction with SGD, which is excessively low, causing training to not converge. To ensure training convergence, we made modifications to the optimization strategy. We switch to the AdamW optimizer with an initial learning rate of 1e-3, which gradually reduce to 1e-5 following a cosine schedule. This adjustment led to the training of NOPS converging successfully. The results, as shown in Tab.16 and Tab.17, indicate that the

Table 13: Detailed results for SemanticPOSS

| Method | Head | Medium | Tail |
|---|---|---|---|
| NOPS | 37.5 | 21.9 | 4.4 |
| Ours | 45.0 | 30.8 | 11.2 |

Table 14: Detailed results for SemanticKITTI

| Method | Head | Medium | Tail |
|---|---|---|---|
| NOPS | 26.1 | 28.3 | 7.4 |
| Ours | 33.8 | 32.2 | 8.5 |

mIoU for known classes improve; however, the mIoU for novel classes decrease as a consequence. Nevertheless, our method continue to outperform NOPS by a significant margin.

## F  ESTIMATE THE NUMBER OF NOVEL CLASSES

We agree that given the number of novel classes (denoted as $|C_u|$) is a valid point for consideration. And it is more realistic to consider unknown $|C_u|$. To estimate $|C_u|$, we extend the classic estimation method Vaze et al. (2022) in NCD to point clouds semantic segmentation. Specifically, we set the candidate range of the total number of categories (seen classes number $<|C_{all}|<$ max classes), and apply Kmeans to cluster the labeled and unlabeled point clouds in the training data, using the representation extracted from a known-class pretrained model. Then, we evaluate the clustering performance of known-class under different $|C_{all}|$, and select $|C_{all}|$ with the highest clustering performance as the estimated $|C_{all}|$. In practice, for computational simplicity, we randomly sample 800,000 points from all scenes to cluster. We conduct experiments on splits 0 of SemanticPOSS, setting max classes to 50, which is a large class number in point cloud semantic segmentation. Consequently, the estimated $|C_u|$ is 3, which is close to the ground truth value (GT is 4).

In addition, we conduct experiments in SemanticPOSS split 0 to compare the results of ours method and previous sotaRiz et al. (2023) with the estimated $|C_u|$. As shown in the table below, our method can still significantly outperform NOPS with estimated $|C_u|$.

In addition, we find that under the estimated $|C_u|$, our method achieves better results than the ground truth $|C_u|$. As shown in Figure 2 in paper, when $C_u = 4$, many plants and buildings are misclassified as the car class (a minor class), resulting in poorer learning for buildings and plants. In contrast, at $C_u = 3$, the model ignores the car class, which is relative small comparing to plants and building, allowing for better learning of buildings and plants, thereby achieving superior results.

Table 15: Comparison of results between Ours and NOPS on various novel classes number, in Split 0 of SemanticPOSS, which has 4 novel classes.

| Method | Building | Car | Ground | Plants | mIoU |
|---|---|---|---|---|---|
| NOPS | 25.54 | 0.00 | 68.15 | 34.12 | 31.95 |
| Ours | 64.05 | 0.00 | 82.22 | 67.63 | **53.47** |

## G  MORE ANALYSIS OF ADAPTIVE REGULARIZATION

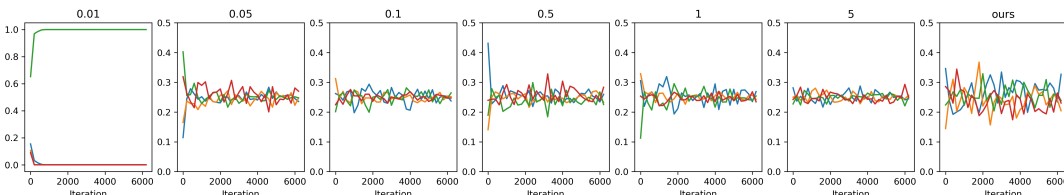

Figure 10: Class distribution for different fixed $\gamma$ during the training.

We visualize the curve showing the changes in the pseudo label distribution before and after adding adaptive regularization, as shown in Fig.9. It is evident that the pseudo label distribution in Baseline+ISL remains consistently uniform during the later stages of training, with each of the four classes

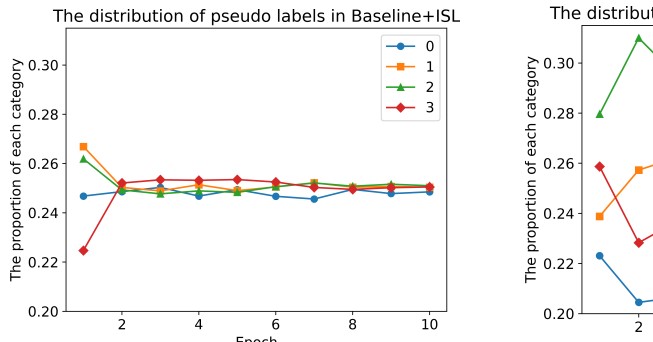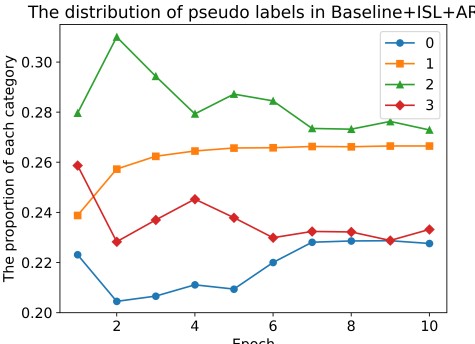

Figure 9: The pseudo label distribution before and after adding adaptive regularization

Table 16: NOPS* denotes NOPS with our training setting.

| Split | Method | bike | build. | car | cone. | fence | grou. | pers. | plants | pole | rider | traf. | trashc. | trunk | Novel | Known | All |
|---|---|---|---|---|---|---|---|---|---|---|---|---|---|---|---|---|---|
| | Full | 45.0 | 83.3 | 52.0 | 36.5 | 46.7 | 77.6 | 68.2 | 77.7 | 36.0 | 58.9 | 30.3 | 4.2 | 14.4 | - | - | 48.5 |
| 0 | NOPS | 35.5 | 30.4 | 1.2 | 13.5 | 24.1 | 69.1 | 44.7 | **42.1** | 19.2 | 47.7 | 24.4 | 8.2 | 21.8 | 35.7 | 26.6 | 29.4 |
| | NOPS* | 46.9 | 16.1 | 4.2 | 35.4 | 47.8 | 54.9 | 67.1 | 37.9 | 36.1 | 62.3 | 28.9 | 1.8 | 20.2 | 28.3 | 38.1 | 35.3 |
| | Ours | 46.5 | **70.3** | **7.6** | 31.1 | 49.4 | **82.2** | 67.1 | 41.7 | 37.5 | 57.5 | 29.8 | 8.4 | 14.1 | **50.4** | 37.9 | 41.8 |
| 1 | NOPS | 29.4 | 71.4 | 28.7 | 12.2 | 3.9 | 78.2 | **56.8** | 74.2 | 18.3 | 38.9 | 23.3 | 13.7 | 23.5 | 30.0 | 38.2 | 36.4 |
| | NOPS* | 21.5 | 83.2 | 49.9 | 29.7 | 20.0 | 77.7 | 29.6 | 77.3 | 37.4 | 58.0 | 26.2 | 3.4 | 15.5 | 23.7 | 45.8 | 40.7 |
| | Ours | **31.5** | 83.2 | 48.7 | 25.4 | **23.9** | 77.3 | 53.1 | 77.1 | 32.5 | 57.3 | 35.0 | 9.3 | 18.0 | **36.2** | 46.4 | 44.0 |
| 2 | NOPS | 37.2 | 71.8 | 29.7 | 14.6 | 28.4 | 77.5 | 52.1 | 73.0 | 11.5 | 47.1 | 0.5 | 10.2 | **14.8** | 9.0 | 44.2 | 36.0 |
| | NOPS* | 44.5 | 83.0 | 51.4 | 25.2 | 47.5 | 77.0 | 66.0 | 76.4 | **23.0** | 59.9 | 4.2 | 8.4 | 4.4 | 10.5 | 53.9 | 43.9 |
| | Ours | 45.3 | 82.8 | 49.8 | 28.4 | 46.3 | 76.7 | 66.2 | 77.2 | 10.9 | 58.4 | **18.6** | 7.3 | 8.2 | 12.6 | 53.8 | 44.3 |
| 3 | NOPS | 38.6 | 70.4 | 30.9 | 0.0 | 29.4 | 76.5 | 56.0 | 71.8 | 17.0 | 31.9 | 26.2 | 1.0 | 22.6 | 10.9 | 43.9 | 36.3 |
| | NOPS* | 44.4 | 83.1 | 46.9 | **0.2** | 43.0 | 77.9 | 65.2 | 78.0 | 34.8 | 45.3 | 32.4 | 1.9 | 14.5 | 15.8 | 52.0 | 43.7 |
| | Ours | 45.5 | 82.9 | 47.7 | 0.0 | 45.1 | 77.8 | 66.3 | 77.7 | 34.3 | **49.1** | 35.6 | **4.0** | 15.3 | **17.7** | 52.8 | 44.7 |

occupying 25%. This indicates that during the latter stages of training, the uniform constraint is too strong, causing the generated pseudo label distribution to lean towards uniformity, which does not align with the actual imbalanced point cloud data, resulting in subpar outcomes. After applying adaptive regularization, we adaptively anneal the uniform constraint based on the KL distance between the pseudo label distribution and a uniform distribution. As shown in the right plot of Fig.9, the curves representing the changes in pseudo label distribution for each class do not tend towards uniformity. This demonstrates that our adaptive regularization, compared to fixed $\gamma$, is more flexible in learning a pseudo label distribution that better aligns with imbalanced point cloud data

To explain how $\gamma$ affects the pseudo-label, we present the class distribution of four classes for different fixed $\gamma$ during the training. Fig.10 shows that: 1. the small $\gamma$ leads to a degenerate solution, 2. enlarging $\gamma$ will gradually approach a uniform distribution, 3. our adaptive $\gamma$ is more flexible and the class distribution is imbalanced.

# H    MORE ANALYSIS OF PROTOTYPE

## H.1    INITIALIZATION AND UPDATING PROCESS

In the beginning, the representation and prototypes are randomly initialized, which is very noisy. However, there are three key factors that guarantee us to gradually improve the representation and prototype. The first one is the learning of seen classes, which improves the representation ability of our model, thus improving the representation of novel classes implicitly. In order to prove that known classes can help the representation of novel classes, we cluster the representation of novel classes obtained from known-class supervised pre-trained model and a randomly initialized model on SemanticPOSS split 0.

Table 17: The novel class discovery results on the SemanticKITTI dataset. 'Full' denotes the results obtained by supervised learning. The gray values are the novel classes in each split.

| Method | bi.cle | b.clst | build. | car | fence | mt.cle | m.clst | oth-g. | oth-v. | park. | pers. | pole | road | side2. | terra. | traff. | truck | trunk | veget. | Novel | Known | All |
|---|---|---|---|---|---|---|---|---|---|---|---|---|---|---|---|---|---|---|---|---|---|---|
| Full | 2.9 | 55.4 | 89.5 | 93.5 | 27.9 | 27.4 | 0.0 | 0.9 | 19.9 | 35.8 | 31.2 | 60.0 | 93.5 | 77.8 | 62.0 | 39.8 | 50.8 | 53.9 | 87.0 | - | - | 47.9 |
| NOPS | 5.6 | 47.8 | 52.7 | 82.6 | 13.8 | 25.6 | 1.4 | 1.7 | 14.5 | 19.8 | 25.9 | 32.1 | **56.7** | 8.1 | **23.8** | 14.3 | 49.4 | 36.2 | **44.2** | 37.1 | 26.5 | 29.3 |
| NOPS* | 7.9 | 55.9 | 46.7 | 89.3 | 24.7 | 27.7 | 0.0 | 1.1 | 22.7 | 25.5 | 33.8 | 57.0 | 43.2 | 17.9 | 21.7 | 39.3 | 61.5 | 50.8 | 23.9 | 30.7 | 35.5 | 34.2 |
| Ours | 5.5 | 51.1 | **74.6** | 92.3 | 29.8 | 22.8 | 0.0 | 0.0 | 23.3 | 24.8 | 27.7 | 59.7 | 41.4 | **22.5** | 23.6 | 39.3 | 43.6 | 51.1 | **66.4** | **45.7** | 33.7 | 36.8 |
| NOPS | 7.4 | 51.2 | 84.5 | 50.9 | 7.3 | 28.9 | 1.8 | 0.0 | 22.2 | 19.4 | 30.4 | 37.6 | 90.1 | 72.2 | 60.8 | 16.8 | 57.3 | **49.3** | 85.1 | 25.4 | 46.2 | 40.7 |
| NOPS* | 2.1 | 52.3 | 89.2 | 52.3 | 6.5 | 27.1 | 0.0 | 0.0 | 18.4 | 17.5 | 33.1 | 59.3 | 90.2 | 77.2 | 61.9 | 39.9 | 53.1 | 19.5 | 86.7 | 19.2 | 49.8 | 41.9 |
| Ours | 3.7 | 57.4 | 89.2 | **56.5** | **17.3** | 20.3 | 0.0 | 0.0 | 20.0 | **30.6** | 34.8 | 60.6 | 93.2 | 77.6 | 62.0 | 38.7 | 56.9 | 39.2 | 86.7 | **28.7** | 50.1 | 44.5 |
| NOPS | 6.7 | 49.2 | 86.4 | 90.8 | 23.7 | 2.7 | 0.6 | 1.9 | **15.5** | 29.5 | 27.9 | **36.4** | 90.3 | 73.4 | 61.2 | **17.8** | 10.3 | 46.2 | 84.3 | 16.5 | 48.0 | 39.7 |
| NOPS* | 4.1 | 55.2 | 89.1 | 93.4 | 29.1 | 0.6 | 0.0 | 0.5 | 2.8 | 33.9 | 30.9 | 32.7 | 93.1 | 77.7 | 60.9 | 0.1 | 32.9 | 52.2 | 86.4 | 13.8 | 50.3 | 40.8 |
| Ours | 3.6 | 54.2 | 88.9 | 93.3 | 28.4 | **10.2** | 0.0 | 0.9 | 9.6 | 33.4 | 32.2 | 36.1 | 92.7 | 77.4 | 62.2 | 10.7 | **34.2** | 51.7 | 86.9 | **20.1** | 50.4 | 42.5 |
| NOPS | 2.3 | 27.8 | 86.0 | 89.9 | 23.1 | 24.5 | **2.9** | 3.1 | 18.2 | 30.1 | **16.3** | 39.9 | 90.7 | 73.5 | 61.0 | 17.4 | 49.8 | 44.0 | 83.2 | 12.4 | 49.0 | 41.2 |
| NOPS* | 2.3 | 16.9 | 89.5 | 93.9 | 28.2 | 27.5 | 0.0 | 0.6 | 25.3 | 34.2 | 3.1 | 60.4 | 93.2 | 77.7 | 61.3 | 38.9 | 67.0 | 54.4 | 86.6 | 5.6 | 55.9 | 45.3 |
| Ours | **2.6** | **32.5** | 88.7 | 93.3 | 28.1 | 24 | 0.1 | 1.0 | 23.7 | 35.6 | 15.3 | 59.8 | 93.2 | 77.6 | 61.4 | 37.8 | 56.6 | 52.1 | 86.7 | **12.6** | 54.6 | 45.8 |

The results indicate that features extracted from known-class pre-trained model exhibit better clustering performance compared to features extracted from a randomly initialized model. The former outperforms the latter by nearly 7% in mIoU for novel classes, demonstrating that known classes can indeed enhance the representation of novel classes. The second one is the view-invariant training, which learns invariant representation for different transformations and promotes the representation directly. Some studies Long et al. (2023a); Zhang et al. (2021) have advanced unsupervised representation learning for point clouds by incorporating transformation invariance. The third one is the utilization of spatial prior, which enforces the point in the same region to be coherent, which may be validated by Fig.2 and 3, and unsupervised clustering Long et al. (2023a); Zhang et al. (2023c).

Those factors gradually improve the representation and prototype, leading to an informative prediction $P$. Then, our adaptive self-labeling algorithm utilizes $P$ and several marginal distribution constraints to generate pseudo-label $Q$. Finally, the $Q$ guides the learning of representation and prototype. In conclusion, the above three factors and our self-labeling learning process ensure our method learn meaningful representation and prototype gradually. Furthermore, we visualize the representation of novel classes during training in Fig.11, showing that as the training time increases, the learned representation gradually becomes better, validating our analysis.

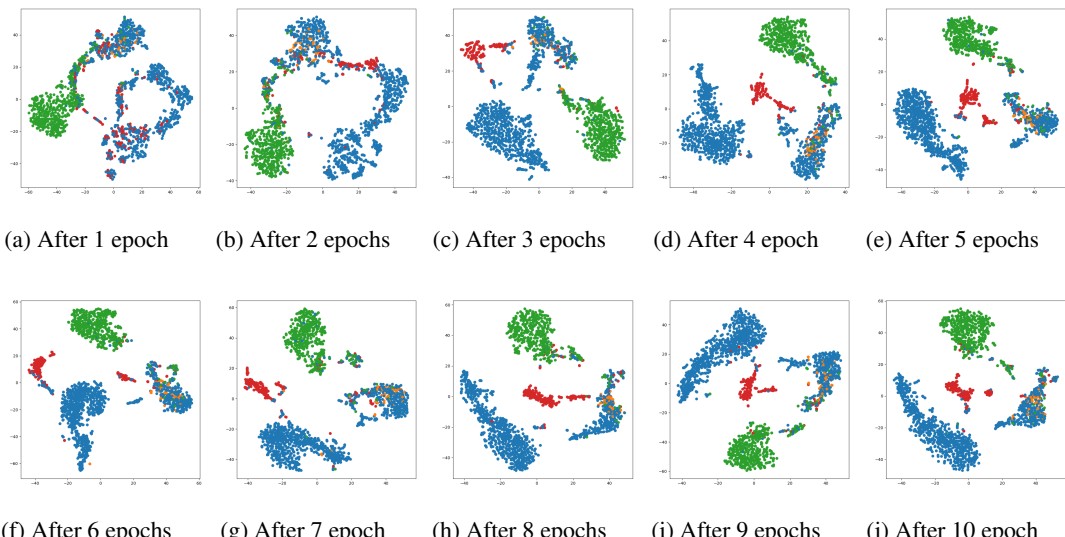

| (a) After 1 epoch | (b) After 2 epochs | (c) After 3 epochs | (d) After 4 epoch | (e) After 5 epochs |
|---|---|---|---|---|

| (f) After 6 epochs | (g) After 7 epoch | (h) After 8 epochs | (i) After 9 epochs | (j) After 10 epoch |
|---|---|---|---|---|

Figure 11: The quality of the representation of unseen classes during training in SemanticPOSS split 0. Blue dots are plants, green dots are the ground, orange are cars, and red are buildings

## H.2 MORE ANALYSIS OF PROTOTYPE SHARING OF REGION-LEVEL LEARNING

We visualize the similarity matrix between the two prototypes, as shown in Figure 12. We observe that the similarity between the two prototypes is low, which validates that having two prototypes leads to disparities in point-wise and region-wise learning directions. In contrast, sharing a prototype avoids this issue.

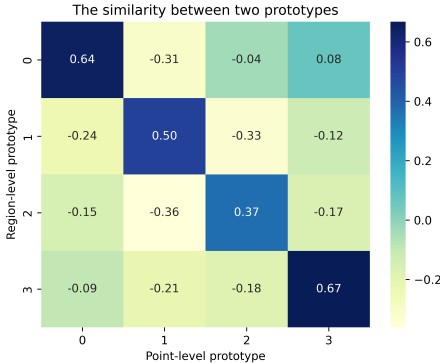

Figure 12: The similarity of two prototypes

## I MORE DETAILS ABOUT AUGMENTATION

Using augmentation to create two views is a well-established technique for learning a transformation-invariant representation, widely employed in novel class discovery literature Fini et al. (2021); Han et al. (2021), and more recently applied in point clouds Riz et al. (2023); Long et al. (2023a); Zhang et al. (2023c). In our comparison, the previous sota methodRiz et al. (2023) also adopts the same augmentation as ours to generate two views. Therefore, our comparison ensures a fair assessment.

To show the effect of augmentation, we conduct ablation on the two views and augmentation.

Table 18: More ablation experiments on SemanticPOSS

| Aug | Two views | ISL+AR+Region | Building | Car | Ground | Plants | mIoU |
|-----|-----------|---------------|----------|-----|--------|--------|------|
|     |           | ✓             | 22.1     | 2.1 | 34.4   | 24.8   | 20.9 |
| ✓   |           | ✓             | 46.3     | 1.9 | 34.5   | 18.6   | 25.3 |
| ✓   | ✓         |               | 21.6     | 2.7 | 76.6   | 26.1   | 31.8 |
| ✓   | ✓         | ✓             | 51.5     | 6.0 | 83.0   | 53.1   | 48.4 |

As Tab.18, From the results above, we draw the following conclusion: 1) Compare column 1 and 2, augmentation on one view has improved by 4.4% in novel classes compared with no augmentation; 2) Compare column 2 and 4, employing two views has improved by 23.1% in novel classes; 3) Compare column 3 and 4, our proposed techniques result in a significant improvement of 17%.

## J MORE ABLATION STUDY

To further analyze the roles of "Region", we supplement ablation experiments on SemanticPOSS using Baseline+ISL+Region. As shown in the Tab.19, "Region" significantly improved performance by 9.1% in split 0 and 3% overall, which is not a minor contribution.

Table 19: More ablation experiments on SemanticPOSS

| ISL | AR | Region | Building | Car | Ground | Plants | Avg | Avg |
|---|---|---|---|---|---|---|---|---|
| | | | 21.6 | 2,7 | 76.6 | 26.1 | 31.8 | 20.9 |
| ✓ | | | 27.6 | 3.1 | 81.2 | 32.1 | 36.0 | 23.9 |
| ✓ | ✓ | | 53.1 | 5.3 | 81.1 | 37.4 | 44.2 | 28.4 |
| ✓ | | ✓ | 41.9 | 9.3 | 83.6 | 45.6 | 45.1 | 26.9 |
| ✓ | ✓ | ✓ | 51.5 | 6.0 | 83.0 | 53.1 | 48.4 | 30.2 |

We conduct additional ablation on split 0 of SemanticKITTI. The results are shown in Tab.20: Similar to SemanticPOSS, each component enhances performance. Specifically, adaptive regularization and region-level learning individually contribute to a 6.6% and 5.0% improvement in mIoU for the model.

Table 20: More ablation experiments on SemanticKITTI split 0

| ISL | AR | Region | Building | Road | Sidewalk | Terrain | Vegetation | Avg |
|---|---|---|---|---|---|---|---|---|
| | | | 46.7 | 43.2 | 17.9 | 21.7 | 23.9 | 30.7 |
| ✓ | | | 57.4 | 32.1 | 25.2 | 18.9 | 37.2 | 34.1 |
| ✓ | ✓ | | 70.8 | 34.7 | 23.2 | 16.8 | 57.9 | 40.7 |
| ✓ | ✓ | ✓ | 74.6 | 41.4 | 22.5 | 23.6 | 66.4 | 45.7 |

# K  MORE DETAILS ABOUT DBSCAN

## K.1  PARAMETERS FOR THE DBSCAN ALGORITHM

DBSCAN is a density-based clustering algorithm that groups points close to each other while marking outliers that exist in low-density regions. DBSCAN has two key parameters: epsilon and min- samples. epsilon represents the maximum distance between two samples for one to be considered as in the neighborhood of the other, while min-samples denotes the minimal number of samples in a region. In our experiments, we set min-samples to be resonable minimal 2, indicating that there must be at least two points in a region. For epsilon, we determine a value of 0.5 based on the proportion of outliers, ensuring that 95% of the point clouds participate in region branch learning. In the following part, we conduct experiments with different epsilon values and analyze the results.

## K.2  VISUAL EXAMPLES OF THE RESULTANT REGIONS

We present visualizations of regions under different epsilon values in Fig.13. As shown in the visualizations, a smaller epsilon results in more outliers and smaller generated regions. Conversely, a higher epsilon leads to fewer outliers and larger generated regions.

## K.3  HOW VARIATIONS IN DBSCAN PARAMETERS MAY IMPACT THE RESULTS

As shown in the Tab.21, we supplement the proportion of outlier points in the 7th column and model training results under different epsilon in the 6th column. To assess the quality of regions, we assign a category label to each region based on the category with the highest point count within the region, with outliers being disregarded, then calculate the mIoU in the 8th column. The results indicate that selecting 0.5 based on the outlier ratio yields satisfactory outcomes. Moreover, fine-tuning epsilon, for instance, setting it to 0.7, leads to improved performance. It is worth noting that the results first increased and then decreased with the increase of epsilon. This is because when epsilon is low, as shown in the visualization, there are more outliers, the generated region is smaller, and less spatial context information is used. When epsilon is higher, the generated region is larger and the Regions mIoU is lower, resulting in noisy region-level representation.

Table 21: Model training results, proportion of outlier points, and region miou under different epsilon. Region mIoU is the mIoU between regions label and ground true. The regions label is composed of each region label, which is the category with the most points in each region. Region mIoU ignores outliers.

| epsilon | Building | Car | Plants | Ground | mIoU | Outlier | Region mIoU |
|---------|----------|-----|--------|--------|------|---------|-------------|
| 0.1 | 41.5 | 1.7 | 45.6 | 80.7 | 42.4 | 45.6% | 97.0 |
| 0.3 | 49.2 | 8.3 | 49.2 | 83.8 | 47.6 | 7.5% | 84.5 |
| 0.5 | 51.5 | 6.0 | 53.1 | 83.0 | 48.4 | 2.5% | 74.8 |
| 0.7 | 65.5 | 9.0 | 61.3 | 78.2 | 53.5 | 1.3% | 64.5 |
| 1 | 49.2 | 8.9 | 55.3 | 82.9 | 49.1 | 0.5% | 44.3 |

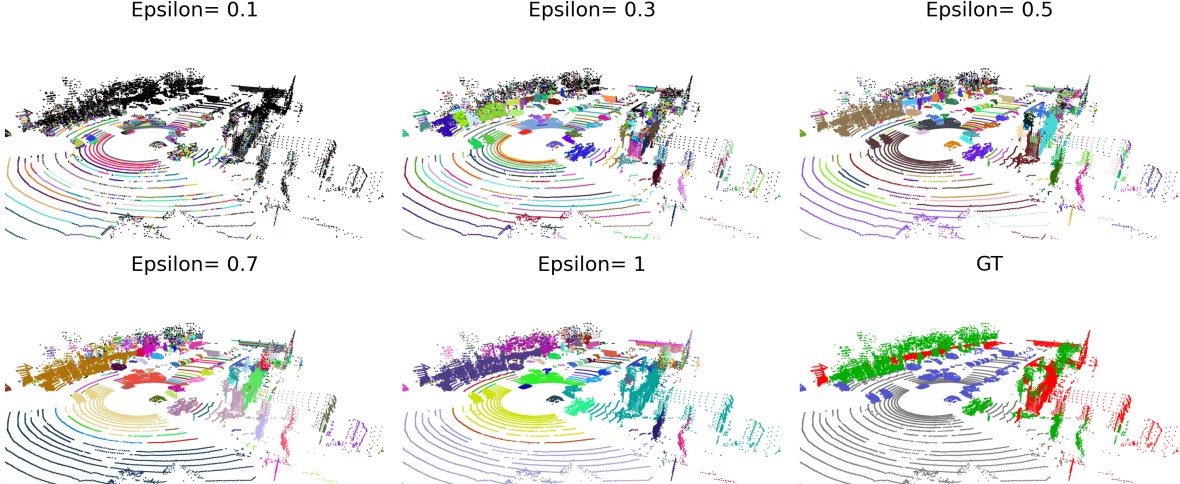

Figure 13: Visualization of regions under different epsilon. For the first five pictures, the black point clouds are outliers, which means that the point does not belong to any region. Other random colors represent a region.

## L MORE VISUALIZATION

To demonstrate the effectiveness of our method, we create a video to compare NOPS with our prediction results, and our method shows a significant improvement over NOPS.

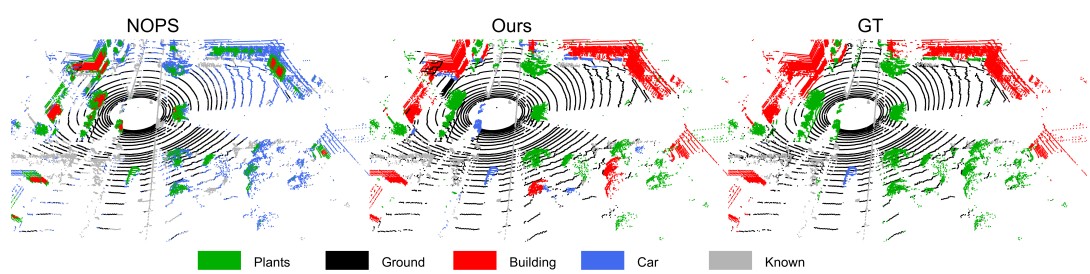

Figure 14: One frame from our video, the dataset being SemanticPOSS split 0.

We also conduct visualization on SemanticKITTI, and the visualizations for different splits are presented below.

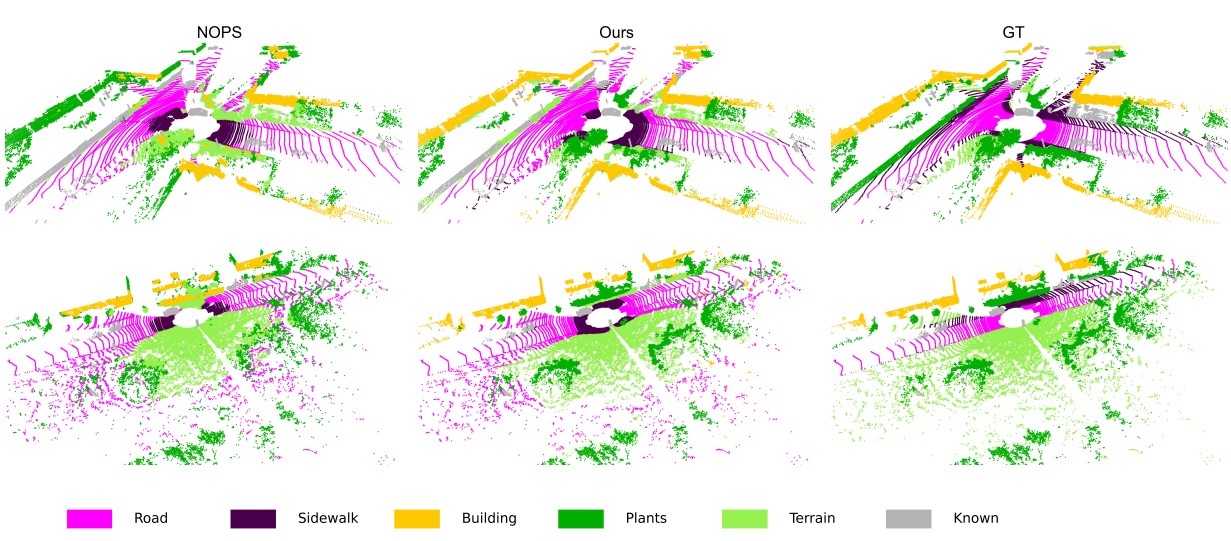

Figure 15: Comparison visualization on SemanticKITTI split 0. Our method exhibits a significant improvement over NOPS.

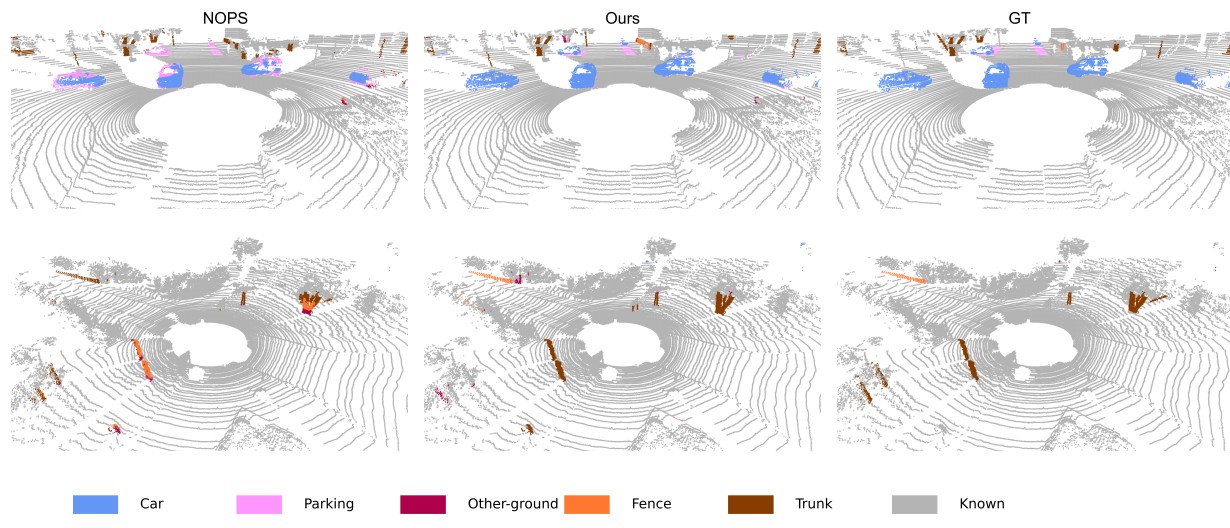

Figure 16: Comparison visualization on SemanticKITTI split 1. Our method exhibits a significant improvement over NOPS.

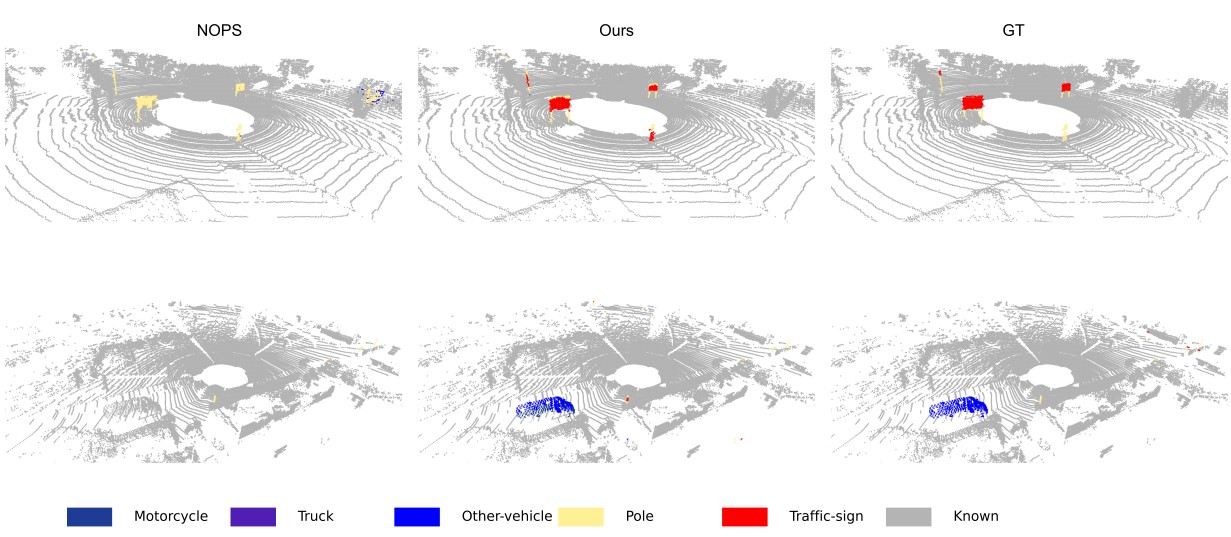

Figure 17: Comparison visualization on SemanticKITTI split 2. Our method exhibits a significant improvement over NOPS.

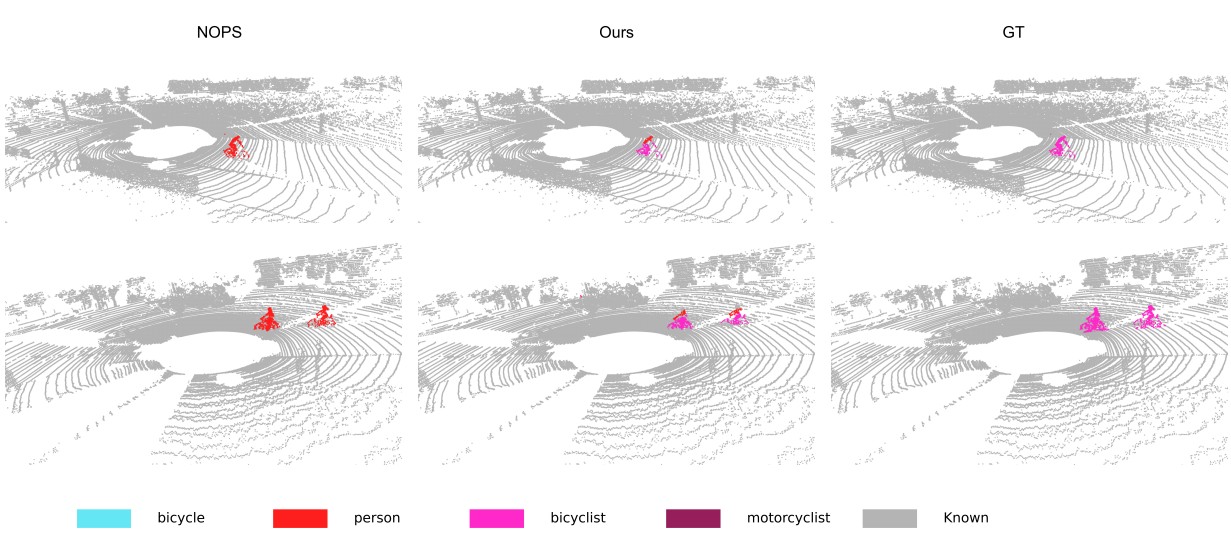

Figure 18: Comparison visualization on SemanticKITTI split 3. Our method exhibits a significant improvement over NOPS.

