# OpenReview forum: "Dual-level Adaptive Self-Labeling for Novel Class Discovery in Point Cloud Segmentation"
_ICLR.cc/2024/Conference — Submitted to ICLR 2024_

### Official Review · Reviewer_Zvms · 2023-10-27

**Soundness:** 2 fair
**Presentation:** 1 poor
**Contribution:** 2 fair
**Rating:** 5
**Confidence:** 3

**Summary:**

This paper effectively addresses the problem of novel class discovery in point cloud segmentation. The proposed method includes a novel self-labeling strategy and a dual-level representation. A regularization technique is also introduced for the self-labeling process. From the experiment, the proposed method is shown to be promising and effective.

**Strengths:**

New methodology for the novel class discovery in point cloud segmentation.
Well-formulated problem.
Promising experimental performance.

**Weaknesses:**

I would say that this paper suffers from a lack of clarity a lot.
The readability is also limited, making readers difficult to get to the point.
The contributions are quite scattered.
There are errors in the presentation.

**Questions:**

There are many questions and problems on this paper.
(1) Motivation is not straightforward. More specifically, the relation between self-labeling and dual-level representation is not clear. It seems that this study addresses two individual problems of the point-wise clustering method. The background of the point-wise clustering should be more informative to improve the readability.
(2) I am not quite sure what the "degenerate solution" means for the point-cloud segmentation. It is also quite unclear why we need a semi-relaxed optimal transport problem definition from the introduction. What is the role of data-dependent annealing in the self-labeling process and dual-level representation? This is also not clear in the introduction.
(3) For the Figure 1, the presentation is quite confusing, and there are even errors. For example, what do different colors represent? Which component represents the novel pseudo-label generation process? Where is the dual-level representation? The arrows among p and y on the right-most part of the figure are quite difficult to understand as well. The "Encoder f_theta" has been occluded...
(4) In the experiment, especially Table 4, I found that the "Region" design only makes a minor contribution to the final performance, while the "dual-representation" is one of the major claimed contributions, making me feel that this paper is quite scattered. There is not a clear focus on tackling a specific problem, making me quite doubtful about the contribution of this paper. I agree that the studied topic is of great importance, but I believe the paper requires thorough refinement before publication.

---

> ### Author Response · Authors · 2023-11-19
> **Response to Reviewer Zvms**
>
> The major concerns of reviewer Zvms are that motivation is not clear and straightforward, and the paper is quite scattered. We respectfully disagree with reviewer Zvms and detail the response for each point in below. We hope our response can address reviewer Zvms's concern.
>
> ## Motivation is not straightforward.
>
> In the second paragraph of the introduction, we have highlighted the shortcomings of point-wise clustering based on equal-size constraints. To reiterate, these include 1) overlooking the inherently imbalanced data distribution and 2) neglecting the rich spatial context information of objects. Those two weaknesses are closely related to novel class discovery for the point cloud segmentation problem and are both important issues. Therefore, we believe our motivation is clear, and we respectfully disagree that solving a problem from two important perspectives is a weakness.
>
> The point-wise clustering means we segment novel classes in the point cloud relying on each individual point, ignoring structured information, such as superpoint or region[3,4]. The existing point-wise clustering methods [3,4] usually employ equal-size constraints on cluster size to avoid degenerate solutions, where all classes are assigned to a single cluster.
>
> ## degenerate solution and semi-relaxed optimal transport problem
>
> In unsupervised clustering, the degenerate solution means we learn a single cluster [1,2] for all classes. Specifically in point cloud segmentation, it means we segment all unlabeled points to a single class. To avoid the degenerate solution, they [1, 3] usually introduce a uniform distribution constraint and formulate the pseudo-label generation problem as an optimal transport problem (Eq 3 in manuscript). **However, such a uniform and equality constraint contradicts with point cloud segmentation problem, where the class distribution is inherently imbalanced, thus generating a noisy pseudo label (Figure 2 in manuscript).** To mitigate this issue, we propose a semi-relaxed optimal transport problem, which relaxes the equality constraint into a KL constraint (Eq 4) and enables us to generate high-quality pseudo-labels (Figure 2 in manuscript). Note that our manuscript has clearly motivated the semi-relaxed optimal transport in the second paragraph, and we have updated the illustration of degenerate solutions for better understanding.
>
> The data-dependent annealing strategy is specifically designed for the self-labeling process and orthogonal with dual-level representation. It adaptively adjusts the regularization strength of KL constraint in semi-relaxed optimal transport problems, enhancing the flexibility of our formulation (see the last sentence of Paragraph 4 in the Introduction).
>
>
> ## Figure 1
>
> We appreciate your suggestion and apologize for any misunderstanding. Please refer to the General Response Part 1 for clarification.

---

> > ### Author Response · Authors · 2023-11-19
> > **Response to Reviewer Zvms**
> >
> > ## ..region .. makes a minor contribution ... this paper is quite scattered...
> >
> > We respectively disagree with Reviewer Zvms that the "Region" design only makes a minor contribution to the final performance. We argue that our main contribution is the novel adaptive self-labeling framework and dual-level representation, which improves 8.2 and 4.2 on split 0, respectively, and 4.5 and 1.8 on all splits. And the improvement of the latter is based on a higher starting point.
> >
> > To further analyze the roles of "Region", we supplement ablation experiments on SemanticPOSS using Baseline+ISL+Region. As shown in the table below, "Region" significantly improved performance by 9.1% in split 0 and 3% overall. We consider "Region" and adaptive regularization to be comparable.
> >
> > |      |      |        |          |      | split 0 |        |      | Overall |
> > | :--: | :--: | :----: | :------: | :--: | :-----: | :----: | :--: | :-----: |
> > | ISL  |  AR  | Region | Building | Car  | Ground  | Plants | Avg  |   Avg   |
> > |      |      |        |   21.6   | 2,7  |  76.6   |  26.1  | 31.8 |  20.9   |
> > |  ✓   |      |        |   27.6   | 3.1  |  81.2   |  32.1  | 36.0 |  23.9   |
> > |  ✓   |  ✓   |        |   53.1   | 5.3  |  81.1   |  37.4  | 44.2 |  28.4   |
> > |  ✓   |      |   ✓    |   41.9   | 9.3  |  83.6   |  45.6  | 45.1 |  26.9   |
> > |  ✓   |  ✓   |   ✓    |   51.5   | 6.0  |  83.0   |  53.1  | 48.4 |  30.2   |
> >
> > We have to reclaim that our two contributions focus on tackling novel class discovery for point cloud segmentation from two important perspectives, which are both well illustrated in the introduction.
> >
> > [1] Yuki M. Asano, Christian Rupprecht, and Andrea Vedaldi. Self-labeling via simultaneous clustering
> > and representation learning. In International Conference on Learning Representations (ICLR),
> > 2020.
> >
> > [2] Caron, Mathilde and Bojanowski, Piotr and Joulin, Armand and Douze, Matthijs. Deep clustering for unsupervised learning of visual features. Proceedings of the European conference on computer vision (ECCV), 2018.
> >
> > [3] Luigi Riz, Cristiano Saltori, Elisa Ricci, and Fabio Poiesi. Novel class discovery for 3d point cloud semantic segmentation. In Proceedings of the IEEE/CVF Conference on Computer Vision and
> > Pattern Recognition, pp. 9393–9402, 2023.
> >
> > [4] Yuyang Zhao, Zhun Zhong, Nicu Sebe, and Gim Hee Lee. Novel class discovery in semantic segmentation. In Proceedings of the IEEE/CVF Conference on Computer Vision and Pattern Recognition, pp. 4340–4349, 2022.

---

> ### Comment · Reviewer_Zvms · 2023-11-20
>
> I appreciate the answers from the authors. Some of my concerns have been addressed, and I can increase my rate. However, I still think this paper requires a focus on a specific problem and a major revision before publication.

---

> > ### Author Response · Authors · 2023-11-20
> >
> > We sincerely appreciate the reviewer's positive feedback. We would like to further clarify that our paper aims to address a specific problem from two crucial perspectives, and we hope the following clarification could address your concern.
> >
> > We aim to tackle the segmentation of novel classes in point clouds, which inherently involve imbalanced and dense prediction tasks. Considering the distinctive nature of this problem, we propose two techniques to mitigate these challenges. Firstly, the novel adaptive self-labeling framework is designed to handle imbalanced clustering, and secondly, the novel dual-level representation addresses the issue of dense prediction.
> >
> > It is important to note that our method addresses a challenging problem from two essential perspectives. We believe that our proposed solution, which integrates those two perspectives, is a strength rather than a weakness.
> >
> > In conclusion, we believe that our method is coherent and specifically focuses on the novel class segmentation in point clouds.

---

### Official Review · Reviewer_cd9p · 2023-10-28

**Soundness:** 3 good
**Presentation:** 3 good
**Contribution:** 3 good
**Rating:** 6
**Confidence:** 3

**Summary:**

This paper studies an interesting problem of novel class discovery in 3D point cloud.
Targeting the weakness of the existing method, this paper proposes two strategies to improve in two directions:
- To improve the ability to cope with class imbalance for novel class discovery, this paper proposes a self-labeling learning procedure that leverages prototypical pseudo-labels. The class distribution of pseudo-labels is regularized under the framework of relaxed Optimal Transport (OT).
- To improve the ability to utilize the spatial relations among points, the authors propose to utilize both point-level and region-level features for pseudo-label generation.

**Strengths:**

The formulation of relaxed OT is interesting and the paper shows how the adaptive regularization strategy is effective in annealing its regularization on class balance/imbalance with both theoretical meaning and empirical studies.

The empirical results are promising compared to previous methods.

This paper is well-written and easy to follow.

**Weaknesses:**

In Fig 1, the method requires 2 views of the same point cloud. Though not quite mentioned, I guess it indicates two independently augmented views of the same point cloud. This is however quite abnormal considering that the pseudo-label is based on prototypes. The paper does not really explain how the augmentation is used alongside all these proposed strategies, which is quite concerning: How  such augmentations affect the method? Does the method really need such augmentations to function? And how does it compare to the method in comparison, eg fairness?

The paper uses heavy notations but most of them are not quite explained and I would recognize some of the notation as unnecessary, especially the overly used sub/superscripts, which hinders the readability of this paper. For example, the notations in 3.1 and 3.2.

Regarding the relaxed OT, the author adds an entropy term to Eq 4 to enable the use of fast scaling algorithms. However, this detail is not explained, regarding how it affects the OT solution and thus pseudo-label generation. In Appendix A, the discussion seems to be more focused on time complexity but not the class implance aspects.
Considering that the authors claim novelty in such formulation of relaxed OT, more discussion and analysis is required, especially regarding how OT captures the class balance/imbalance and its impact.

Besides, since the authors emphasize the dynamic nature of weight r, which seems to be a key difference to previous methods, it's nice to see how this relaxed OT reduces to the original OT for a balanced class by controlling the r, if possible.

Also, how r affects the pseudo-label deserves more discussion and analysis. For example, a visualization on pseudo-labels of larger/smaller r, or theoretical analysis, would be desired.

**Questions:**

see weaknesses

---

> ### Author Response · Authors · 2023-11-19
> **Response to Reviewer cd9p**
>
> We appreciate your valuable feedback and hope the following response could address your concern.
>
> ## Augmentation
>
> Thanks to your suggestion regarding our two augmented views, we indeed adopt augmentation to generate two views for learning transformation-invariant representation. Using augmentation to create two views is a well-established technique for learning a transformation-invariant representation, widely employed in novel class discovery literature [2,3], and more recently applied in point clouds[1,4,5]. In our comparison, the previous sota method [1] also adopts the same augmentation as ours to generate two views. Therefore, our comparison ensures a fair assessment.
>
> In our method, we first generate two views of the same point cloud by different augmentation, including naive rotation and scale, which are adopted from previous work [1] and have been illustrated in Implementation Details. Then, we generate point-level and region-level imbalanced pseudo-labels through adaptive self-labeling on two views in parallel. Subsequently, we swap the pseudo-labels generated from the two views to learn transformation invariant representation.
>
> To show the effect of augmentation, we conduct ablation on the two views and augmentation.
>
> | Aug  | Two views | ISL+AR+Region | Building | Car  | Ground | Plants | mIoU |
> | ---- | --------- | ------------- | :------: | :--: | :----: | :----: | :--: |
> |      |           | √             |   22.1   | 2.1  |  34.4  |  24.8  | 20.9 |
> | √    |           | √             |   46.3   | 1.9  |  34.5  |  18.6  | 25.3 |
> | √    | √         |               |   21.6   | 2.7  |  76.6  |  26.1  | 31.8 |
> | √    | √         | √             |   51.5   | 6.0  |  83.0  |  53.1  | 48.4 |
>
> From the results above, we draw the following conclusion:
> * Compare column 1 and 2, augmentation on one view has improved by 4.4% in novel classes compared with no augmentation.
> * Compare column 2 and 4, employing two views has improved by 23.1% in novel classes.
> * Compare column 3 and 4, our proposed techniques result in a significant improvement of 16.6%.
>
>
>
> [1] Luigi Riz, Cristiano Saltori, Elisa Ricci, and Fabio Poiesi. Novel class discovery for 3d point cloud semantic segmentation. In Proceedings of the IEEE/CVF Conference on Computer Vision and Pattern Recognition, pp. 9393–9402, 2023.
>
> [2] Enrico Fini, Enver Sangineto, Stephane Lathuiliere, Zhun Zhong, Moin Nabi, and Elisa Ricci. `
> A unified objective for novel class discovery. In Proceedings of the IEEE/CVF International
> Conference on Computer Vision, pp. 9284–9292, 2021.
>
> [3] Kai Han, Sylvestre-Alvise Rebuffi, Sebastien Ehrhardt, Andrea Vedaldi, and Andrew Zisserman. Autonovel: Automatically discovering and learning novel visual categories. IEEE Transactions on Pattern Analysis and Machine Intelligence, 2021.
>
> [4] Fuchen Long, Zhaofan Yao, Ting abd Qiu, Lusong Li, and Tao Mei. Pointclustering: Unsupervised point cloud pre-training using transformation invariance in clustering. In CVPR, 2023.
>
> [5] Zihui Zhang, Bo Yang, Bing Wang, and Bo Li. Growsp: Unsupervised semantic segmentation of
> 3d point clouds. In Proceedings of the IEEE/CVF Conference on Computer Vision and Pattern
> Recognition, pp. 17619–17629, 2023c.
>
> ## Heavy notation
>
> We appreciate your suggestion. We have simplified most notation in Section 3.1 for better readability but kept the sub/superscripts in Section 3.2 for clarity. The superscripts are usually used to distinguish the point and region, the subscripts are usually the index.

---

> > ### Author Response · Authors · 2023-11-19
> > **Response to Reviewer cd9p**
> >
> > ## More details about relaxed OT
> >
> > ### Entropy term
> >
> > The introduction of the entropy term enables us to use fast-scaling algorithms, which has been proved by [1] and becomes a typical solution for the application of OT. Without the entropy term, the computation complexity of the original OT is nearly $O(n^3logn)$  [1], which is unaffordable for a realistic problem.  Due to the entropy term is not our contribution, we do not analyze too much in our manuscript and refer the reader to [1] for more details.
> >
> > ### Class Imbalance
> >
> > Thanks for your advice, we would like to clarify how our method captures the class imbalance, and have updated the following analysis to Appendix A.
> >
> >
> > As discussed in the General Response Part 2, the representation and prototype will gradually bootstrap. Therefore, the imbalanced property of the dataset will be reflected in the model's prediction P. OT generates pseudo labels (Q) based on P and several distribution constraints. Unlike typical OT, which imposes two equality constraints and enforces a uniform distribution, our relaxed OT utilizes a relaxed KL constraint on cluster size.
> >
> > In the optimization of our relaxed OT, the optimal Q for the <Q, -log P>  is to assign Q based on the largest prediction in P,  thus capturing the imbalanced property of P, and the KL constraints enforce the marginal distribution of Q not far away from the prior uniform distribution, avoiding degenerate solution while providing some flexibility. Consequently, the optimal Q in our relaxed OT should take into account both the inherent imbalanced distribution of classes in P and a prior uniform distribution,  thus generating pseudo labels reflecting the imbalanced characteristics of P.
> >
> > [1] Marco Cuturi. Sinkhorn distances: Lightspeed computation of optimal transport. Advances in neural information processing systems, 26, 2013.
> >
> > ## More analysis on $\gamma$
> >
> > As analysis above, ideally, the relaxed OT can generate imbalanced pseudo labels, due to imbalanced property inherent in P and relaxed KL constraint. However, the constraint of KL should be well-tuned. A larger KL will enforce the distribution close to uniform, while a small KL tends to learn a degenerate solution. Therefore, in our manuscript, we argue a dynamic KL that adjusts its strength based on the model learning process is needed.
> >
> > In Appendix F, we present the class distribution of four classes for different fixed $\gamma$ during the training. It shows that: 1）the small $\gamma$ leads to a degenerate solution, 2) enlarging $\gamma$ will gradually approach a uniform distribution, 3) our adaptive $\gamma$ is more flexible and the class distribution is imbalanced.

---

> ### Author Response · Authors · 2023-11-23
> **Additional Response to Reviewer cd9p**
>
> Thank you for your valuable feedback on our submission. We appreciate your detailed feedback and appraisal of our work, which we have taken into careful consideration in our rebuttal response. As the rebuttal process is coming to an end, we would be grateful if you could acknowledge receipt of our responses and let us know if they address your concerns. We remain eager to engage in any further discussions.

---

### Official Review · Reviewer_nnmw · 2023-10-29

**Soundness:** 3 good
**Presentation:** 2 fair
**Contribution:** 2 fair
**Rating:** 5
**Confidence:** 3

**Summary:**

This work presents a novel method for novel class discovery in point cloud segmentation. Specifically, the authors proposed a self-labeling strategy for addressing imbalanced classes and introduced a dual-level representation to enhance regional consistency. The experiments demonstrate that this method leads to a significant improvement in performance.

**Strengths:**

1. The proposed method is well-motivated, addressing the issues of imbalanced classes and regional consistency.
2. The experimental results are comprehensive and meticulously detailed. The impact of different settings and various components of the method is considered and discussed.

**Weaknesses:**

1. The paper's content lacks a smooth organization and its order creates confusion regarding the key method modules. Figure 1 is incomplete, with unexplained symbols.
2. The analysis of the experimental results is not very sufficient.

**Questions:**

1. Does the parameters of DBSCAN have an impact on the final result?
2. In the ablation results, the application of region-level learning results in a decrease in the IoU value for 'Building.' Why does this phenomenon occur, and is this method more beneficial for smaller objects?

---

> ### Author Response · Authors · 2023-11-19
> **Response to Reviewer nnmw**
>
> We appreciate your valuable feedback and hope the following response could address your concern.
>
> ## Content lacks a smooth organization .. and figure 1 is incomplete ..
>
> We appreciate your suggestion and have carefully considered your suggestions. However, we believe that our organizational structure is reasonable, as we employ a standard format. We present an overview of our method in Section 3.1, followed by a description of our network architecture, encompassing dual-level representation and a prototype-based classifier. Subsequently, we outline the overall self-labeling framework and provide details on imbalanced pseudo-label generation and the adaptive regularization strategy. To improve clarity, we have reorganized Section 3.1 to ensure coherence. It's notable that other reviewers (cd9p, 74h9) have also praised that "it is well-written and easy to follow".  We would appreciate it if you could specify which part lacks a smooth organization.
>
> We appreciate your advice on Figure 1, and please refer to the General Response Part 1 for details.
>
> ## Analysis of the experimental results is not sufficient
>
> ###  Hpyer-parameters of DBSCAN
>
> Please refer to General Response.
>
>
> ### Decrease in the IoU value for "Building"
>
> **Explanation of the decrease of "Building"**
>
> We appreciate your feedback. As depicted in Table 4, the inclusion of region-level features results in only a relatively small decrease in the "Building" class (53.1% -> 51.5%), while the average mIoU increases by 4.2%. We note that the performance of "Building" is improved on the training data after incorporating region-level learning, as shown in Figure 3. Therefore, we speculate that the marginal decrease in the building class may be due to the model overfitting to the training set.
>
> Distinct from other classes, such as plants, cars, and grounds, "Building" is positioned farther away from the sensor, leading to a sparse spatial distribution and relatively small regions generated from DBSCAN. Therefore, we suspect that this phenomenon may arise from the prototypes capturing the bias of the relatively small region associated with the "Building" in the training data, which is inconsistent with the test data. This issue can be addressed by introducing additional regularization. For instance, as demonstrated in the table in General Response 2, configuring DBSCAN to generate relatively larger regions (e.g., with epsilon=0.7) significantly increases the results for the building class.
>
> **Is this method more beneficial for smaller objects?**
>
> Our method is more beneficial to dense objects instead of smaller objects. Specifically, we utilize DBSCAN, which is a density-based clustering algorithm, to generate regions for point clouds with relatively high density. Therefore, our method is beneficial for objects relative to density, rather than size.

---

> ### Author Response · Authors · 2023-11-23
> **Additional Response to Reviewer nnmw**
>
> Thank you so much for taking the time to review our submission. We appreciate your detailed feedback and appraisal of our work, which we have taken into careful consideration in our rebuttal response. As the rebuttal process is coming to an end, we would be grateful if you could acknowledge receipt of our responses and let us know if they address your concerns. We remain eager to engage in any further discussions.

---

### Official Review · Reviewer_74h9 · 2023-10-29

**Soundness:** 3 good
**Presentation:** 3 good
**Contribution:** 2 fair
**Rating:** 6
**Confidence:** 4

**Summary:**

This paper investigates the task of novel class discovery in point cloud segmentation. It addresses two major issues that were present in the previous work NOPS (Riz et al. 2023): the equal class-size constraint and the omission of spatial context information during point-wise clustering. The authors introduce an adaptive self-labeling method. This method relaxes the optimal transport problem by transforming it into a semi-relaxed optimal transport (OT) problem with an annealing-like regularization strategy. Moreover, the proposed approach includes a region-level branch by clustering points into regions and generating region-level features via pooling, complementing the point-level features for prediction. The proposed method is evaluated on two outdoor datasets, and it demonstrates impressive performance compared to the baseline method NOPS.

**Strengths:**

1. The proposed method is technically sound and is well-motivated.
2. Notably, the results on the SemanticPOSS dataset are remarkable and demonstrate the effectiveness of the proposed approach.
3. This paper includes comprehensive ablation studies that thoroughly validate the impact of various components within the framework. This provides a strong basis for the proposed method's effectiveness.

**Weaknesses:**

1. The utility of the novel class discovery setting, where the number of novel classes is predetermined, is a valid point for consideration. In an open-world scenario, the number of novel classes is often dynamic and not known in advance.
2. Adding more specific details about the clustering process would provide a better understanding. This could include parameters for the DBSCAN algorithm, visual examples of the resultant regions, and how variations in DBSCAN parameters may impact the results. Additionally, clarification on whether "K" is a fixed value for different scenes and how these "K" regions are generated during clustering is needed.
3. Elaborating on the initialization and updating process for class prototypes would be beneficial to better understand the methodology.
4. Discussing the applicability of the proposed framework in indoor scenarios, where there are typically more novel classes to discover, would enhance this paper's practical relevance.
5. Conducting an ablation study on a split of the SemanticKITTI dataset, which is known to be more challenging than SemanticPOSS, would strengthen this paper's findings.


A minor point: In Table 3, adding a column that indicates the split index, similar to Table 1, would improve clarity.

**Questions:**

Please refer to the comments in the weaknesses section.

---

> ### Author Response · Authors · 2023-11-19
> **Response to Reviewer 74h9**
>
> We appreciate Reviewer 74h9 valuable feedback and respond to the concerns in the following.
>
> ## Unknown the number of novel classes.
>
> We agree that the utility of the novel class discovery setting is a valid point for consideration. In our manuscript, we divide novel class discovery in point clouds into two sub-problems. The first is to estimate the number of novel classes. The second problem is to discover novel classes after determining the number of novel classes, which is challenging due to imbalanced class distribution. Additionally, to ensure a fair comparison with other methods [2], we use the number of novel classes as a prior and focus on the second problem.
>
> And it is more realistic to consider unknown $|C_u|$. To estimate $|C_u|$, we extend the classic estimation method [1] in NCD to point cloud semantic segmentation. Specifically, we set the candidate range of the total number of categories (seen classes number $\textless |C_{all}|\textless$ max classes), and apply Kmeans to cluster the labeled and unlabeled point clouds in the training data, using the representation extracted from a known-class pre-trained model. Then, we evaluate the clustering performance of known classes under different $|C_{all}|$, and select $|C_{all}|$  with the highest clustering performance as the estimated $|C_{all}|$. In practice, for computational simplicity, we randomly sample 800,000 points from all scenes to cluster. We conduct experiments on splits 0 of SemanticPOSS, setting max classes to 50, which is a large class number in point cloud semantic segmentation. Consequently, the estimated $|C_u|$ is 3, which is close to the ground truth value (GT is 4).
>
> In addition, we conduct experiments in SemanticPOSS split 0 to compare the results of our method and previous sota[2] when the estimated value deviates from the true value. As shown in the table below, our method can still significantly outperform NOPS with estimated $|C_u|$.
>
> |      | split 0  |      |        |        |           |
> | ---- | -------- | ---- | ------ | ------ | --------- |
> |      | Building | Car  | Ground | Plants | mIoU      |
> | NOPS | 25.54    | 0.00 | 68.15  | 34.12  | 31.95     |
> | Ours | 64.05    | 0.00 | 82.22  | 67.63  | **53.47** |
>
> In conclusion, our sub-problem division is reasonable, and our method shows significant improvement compared to the previous sota, even when the estimated novel class deviates slightly from the real value.
>
> In addition, we find that under the estimated $|C_u|$, our method achieves better results than the ground truth $|C_u|=4$. As shown in Figure 2 in the paper, when $C_u=4$, many plants and buildings are misclassified as the car class (a minor class), resulting in poorer learning for buildings and plants. In contrast, at $C_u=3$, the model ignores the car class, which is relatively small compared to plants and buildings, allowing for better learning of buildings and plants, thereby achieving superior results.
>
> [1] Sagar Vaze, Kai Han, Andrea Vedaldi, and Andrew Zisserman. Generalized category discovery. In Proceedings of the IEEE/CVF Conference on Computer Vision and Pattern Recognition, pp. 7492–7501, 2022.
>
> [2] Luigi Riz, Cristiano Saltori, Elisa Ricci, and Fabio Poiesi. Novel class discovery for 3d point cloud semantic segmentation. In Proceedings of the IEEE/CVF Conference on Computer Vision and Pattern Recognition, pp. 9393–9402, 2023.
>
>
> ## Parameters of DBSCAN
>
> Please refer to the General Response Part 1.
>
>
> ##  Initialization and updating process for class prototypes
>
> Please refer to the General Response Part 2.
>
> ## Discussing our framework in indoor scenarios
>
> Our framework is general and can be applied to indoor scenarios with little modification. The key difference between indoor scenarios and outdoor scenarios is that Indoor scenarios 1) have more classes; 2) are more denser; 3) include additional RGB information.
>
> As for more novel classes, our results in Table 2 show we can achieve satisfactory results compared to NOPS on harder splits, where half (6 or 7) classes are novel.  As for more dense points, we have to adjust the DBSCAN to generate appropriate regions. For the additional RGB information, we can utilize it to enhance the quality of regions and adjust the backbone to process the input RGB information.
>
> In conclusion, our method can be relatively easily extended to indoor scenarios.

---

> ### Author Response · Authors · 2023-11-19
> **Response to Reviewer 74h9**
>
> ## Ablation study on SemanticKITTI
>
> We appreciate your suggestion and conduct additional ablation on split 0 of SemanticKITTI. The results are shown below:
>
> |      |      |        |          |      | split 0  |         |            |      |
> | :--: | :--: | :----: | :------: | :--: | :------: | :-----: | :--------: | :--: |
> | ISL  |  AR  | Region | Building | Road | Sidewalk | Terrain | Vegetation | Avg  |
> |      |      |        |   46.7   | 43.2 |   17.9   |  21.7   |    23.9    | 30.7 |
> |  ✓   |      |        |   57.4   | 32.1 |   25.2   |  18.9   |    37.2    | 34.1 |
> |  ✓   |  ✓   |        |   70.8   | 34.7 |   23.2   |  16.8   |    57.9    | 40.7 |
> |  ✓   |  ✓   |   ✓    |   74.6   | 41.4 |   22.5   |  23.6   |    66.4    | 45.7 |
>
> Similar to SemanticPOSS, each component enhances performance. Specifically, adaptive regularization and region-level learning individually contribute to a 6.6% and 5.0% improvement in mIoU for the model.
>
>
>
> ## Minor Point
>
> We appreciate your advice. But the table is too wide to insert another column. To make it clear, we add an explanation of the split index to the caption.

---

> > ### Comment · Reviewer_74h9 · 2023-11-22
> > **Official Comment by Reviewer 74h9**
> >
> > Thank the authors for their thorough response, the clarifications provided, and the additional ablation results. My concerns are mostly addressed, and I would like to keep my initial rating.

---

### Author Response · Authors · 2023-11-19
**General Response 1**

We express our gratitude for the valuable feedback from the reviewers and are encouraged by the positive remarks such as "The proposed method is technically sound and is well-motivated" (Reviewer 74h9), "The proposed method is well-motivated" (Reviewer nnmw), "The formulation of relaxed OT is interesting" (Reviewer cd9p), and "Well-formulated problem. Promising experimental performance" (Reviewer Zvms). In the following, we address common concerns and individual concerns separately.

## Parameters of DBSCAN

Thanks to reviewer 74h9 and nnmw, we clarify the following details for DBSCAN:

1. **Parameters for the DBSCAN algorithm**

   DBSCAN is a density-based clustering algorithm that groups points close to each other while marking outliers that exist in low-density regions. DBSCAN has two key parameters: epsilon and min_samples. epsilon represents the maximum distance between two samples for one to be considered as in the neighborhood of the other, while min_samples denotes the minimal number of samples in a region. In our experiments, we set min_samples to be reasonable minimal 2, indicating that there must be at least two points in a region. For epsilon, we determine a value of 0.5 based on the proportion of outliers, ensuring that 95% of the point clouds participate in region branch learning. In the following part, we conduct experiments with different epsilon values and analyze the results.

2. **Visual examples of the resultant regions**

   We present visualizations of regions under different epsilon values in Appendix K. As shown in the visualizations, smaller epsilon results in more outliers and smaller generated regions. Conversely, a higher epsilon leads to fewer outliers and larger generated regions.

3. **How variations in DBSCAN parameters may impact the results**

   As shown in the table below, we supplement the proportion of outlier points in the 7th column and model training results under different epsilon in the 6th column. To assess the quality of regions, we assign a category label to each region based on the category with the highest point count within the region, with outliers being disregarded, and then calculate the mIoU in the 8th column. The results indicate that selecting 0.5 based on the outlier ratio yields satisfactory outcomes. Moreover, fine-tuning epsilon, for instance, setting it to 0.7, leads to improved performance. It is worth noting that the results first increased and then decreased with the increase of epsilon. This is because when epsilon is low, as shown in the visualization, there are more outliers, the generated region is smaller, and less spatial context information is used. When epsilon is higher, the generated region is larger and the Regions mIoU is lower, resulting in noisy region-level representation.

| epsilon | Building | Car  | Plants | Ground | mIoU  | Outlier | Regions mIoU |
| ------- | :------: | :--: | :----: | :----: | :---: | :-----: | :----------: |
| 0.1     |  41.57   | 1.74 | 45.60  | 80.75  | 42.42 |  45.6%  |     97.0     |
| 0.3     |  49.22   | 8.33 | 49.22  | 83.86  | 47.66 |  7.5%   |     84.5     |
| 0.5     |  51.50   | 6.00 | 53.10  | 83.00  | 48.40 |  2.5%   |     74.8     |
| 0.7     |  65.51   | 9.04 | 61.37  | 78.27  | 53.55 |  1.3%   |     64.5     |
| 1       |  49.20   | 8.97 | 55.34  | 82.95  | 49.12 |  0.5%   |     44.3     |


## Figure 1

We have updated Figure 1 for a better understanding. Different colors represent known and novel classes. And we have added legends to describe the meaning of arrows. Specifically, the arrow from p to y in the rightmost part denotes the novel pseudo-label generation process, which is detailed in Section 3.3. The top and bottom parts represent the region-level branch, and the middle part represents the point-level branch.

---

> ### Author Response · Authors · 2023-11-19
> **General Response Part 2**
>
> ## The learning of representation and prototypes
>
> We appreciate the advice from the reviewers and further elaborate on the learning of representation and prototypes. In the beginning, the representation and prototypes are randomly initialized, which is very noisy. However, there are three key factors that guarantee us to gradually improve the representation and prototype. The first one is the learning of seen classes, which improves the representation ability of our model, thus improving the representation of novel classes implicitly.  In order to prove that known classes can help the representation of novel classes, we cluster the representation of novel classes obtained from the known-class supervised pre-trained model and a randomly initialized model on SemanticPOSS split 0.
>
> |                         | Building | Car  | Plants | Ground | mIoU      |
> | ----------------------- | -------- | ---- | ------ | ------ | --------- |
> | Random initialization   | 11.15    | 2.28 | 28.63  | 30.07  | 18.03     |
> | Known-class pre-trained | 26.48    | 3.07 | 27.18  | 43.71  | **25.11** |
>
> The results indicate that features extracted from the known-class pre-trained model exhibit better clustering performance compared to features extracted from a randomly initialized model. The former outperforms the latter by nearly 7% in mIoU for novel classes, demonstrating that known classes can indeed enhance the representation of novel classes.
>
> The second one is the view-invariant training, which learns invariant representation for different transformations and promotes the representation directly. Some studies [3, 4] have advanced unsupervised representation learning for point clouds by incorporating transformation invariance.
>
> The third one is the utilization of spatial prior, which enforces the point in the same region to be coherent, which may be validated by Figures 2 and 3, and unsupervised clustering [1, 2].
>
> Those factors gradually improve the representation and prototypes, leading to an informative prediction $P$. Then, our adaptive self-labeling algorithm utilizes $P$ and several marginal distribution constraints to generate pseudo-label $Q$. Finally, the $Q$ guides the learning of representation and prototype. In conclusion,  the above three factors and our self-labeling learning process ensure our method learns meaningful representation and prototype gradually. Furthermore, we visualize the representation of novel classes during training in Appendix G.1, showing that as the training time increases, the learned representation gradually becomes better, validating our analysis.
>
> [1] Fuchen Long, Zhaofan Yao, Ting abd Qiu, Lusong Li, and Tao Mei. Pointclustering: Unsupervised point cloud pre-training using transformation invariance in clustering. In CVPR, 2023.
>
> [2] Zihui Zhang, Bo Yang, Bing Wang, and Bo Li. Growsp: Unsupervised semantic segmentation of 3d point clouds. In Proceedings of the IEEE/CVF Conference on Computer Vision and Pattern Recognition, pp. 17619–17629, 2023c.
>
> [3] Saining Xie, Jiatao Gu, Demi Guo, Charles R Qi, Leonidas Guibas, and Or Litany. Pointcontrast: Unsupervised pretraining for 3d point cloud understanding. In ECCV, 2020. 3
>
> [4] Zhang Z., Girdhar R., Joulin A., Misra I. Self-supervised pretraining of 3d features on any point-cloud. In: ICCV, 2021

---

### Meta-Review · Area_Chair_4e2L · 2024-01-07

**Metareview:**

Summary:
This paper explores novel class discovery in point cloud segmentation, addressing issues in previous work, including equal class-size constraints and the lack of spatial context information during point-wise clustering. The proposed method involves self-labeling to handle class imbalance and introduces a dual-level representation for improved regional consistency. Evaluated on outdoor datasets, the method outperforms baseline approaches, effectively tackling the challenges in novel class discovery within 3D point cloud segmentation.

Strengths:
The paper introduces a technically sound and well-motivated method for novel class discovery in point cloud segmentation. The formulation of relaxed optimal transport and the adaptive regularization strategy is interesting. Results on the benchmark datasets demonstrate the method's effectiveness compared to previous approaches. The paper includes comprehensive ablation studies, discussing the impact of different settings and various components, presenting promising experimental performance in the field of point cloud segmentation.

Weaknesses:
The paper has several weaknesses that may need more consideration. Firstly, the novel class discovery setting with a predetermined number of novel classes may need more thinking and exploration, especially in dynamic, open-world scenarios where the number of novel classes is unknown in advance. The paper could benefit from providing more clarification and specific details about the clustering process, initialization and updating processes for class prototypes. Moreover, the paper is criticized by reviewers for lacking clarity, limited readability, scattered contributions, and errors in presentations, which collectively hinder readers from grasping the key points effectively.

**Justification For Why Not Higher Score:**

Although some concerns are well addressed by author's responses, I respectively agree with Reviewer Zvms that a major revision may still be needed before publication.

**Justification For Why Not Lower Score:**

N/A

---

### Decision · Program_Chairs · 2024-01-16

Reject